# Conditional Latent Space Molecular Scaffold Optimization for Accelerated Molecular Design

## Abstract

The rapid discovery of new chemical compounds is essential for advancing global health and developing treatments. While generative models show promise in creating novel molecules, challenges remain in ensuring the real-world applicability of these molecules and finding such molecules efficiently. To address this, we introduce Conditional Latent Space Molecular Scaffold Optimization (CLaSMO), which combines a Conditional Variational Autoencoder (CVAE) with Latent Space Bayesian Optimization (LSBO) to modify molecules strategically while maintaining similarity to the original input. Our LSBO setting improves the sample-efficiency of our optimization, and our modification approach helps us to obtain molecules with higher chances of real-world applicability. CLaSMO explores substructures of molecules in a sample-efficient manner by performing BO in the latent space of a CVAE conditioned on the atomic environment of the molecule to be optimized. Our experiments across 22 diverse optimization tasks reveal that CLaSMO efficiently enhances target properties with minimal substructure modifications, delivers notable sample-efficiency—an essential factor in resource-constrained real-world scenarios—and achieves state-of-the-art results while utilizing a smaller model and dataset compared to existing methods. We also provide an open-source web application that enables chemical experts to apply CLaSMO in a Human-in-the-Loop setting.

## 1    Introduction

The accelerated discovery of chemical compounds represents a crucial challenge with the potential to revolutionize global health, offering new ways to combat diseases and viruses. The ability to efficiently discover and develop new chemical compounds could lead to groundbreaking treatments and therapies, addressing some of the most pressing health issues of our time. As the importance of this field grows, so too does the research focused on finding effective solutions. Over the past few years, artificial intelligence (AI) has emerged as a powerful tool in this endeavor. The combination of increased computational power and advancements in generative modeling has brought us closer than ever to achieving significant breakthroughs in accelerated discovery.

Generative models offer various approaches to exploring and creating new chemical compounds. A common strategy involves training a generative model on a comprehensive database of chemical compounds. Once trained, the model can generate entirely new compounds (Gómez-Bombarelli et al., 2018; Tripp et al., 2020; Griffiths & Hernández-Lobato, 2020; Boyar & Takeuchi, 2024; Grosnit et al., 2021; Boyar et al., 2024). This approach opens the door to discovering novel molecules that are unlike any currently known, potentially revealing a vast, untapped universe of chemical possibilities. However, while the generation of these novel molecules is exciting, their practical applicability is often constrained. The challenges lie in synthesizing these novel molecules and in the limited understanding of their properties, which makes it difficult for domain experts to assess their viability (Lim et al., 2020).

To address these challenges, another strategy for designing new molecules focuses on modifying/editing existing compounds using generative models (Bradshaw et al., 2019; Lim et al., 2020), reinforcement learning (Gottipati et al., 2020), genetic algorithms (Jensen, 2019), or from the domain experts themselves by trial

Figure 1: An example of the CLaSMO framework updating a scaffold for QED optimization. CLaSMO identifies optimal regions in the CVAE latent space and selects bonding points on the scaffold. Chemical features from these points are used to guide the generation of substructures, which are then integrated into the scaffold (A, B, C, D) through small, targeted modifications to improve molecular properties.

and error (Bemis & Murcko, 1996; Schreiber, 2000; Welsch et al., 2010). These approaches are more likely to produce synthesizable molecules because the base molecule is known to exist and, therefore, can be more tractable for real-world applications. However, even with this strategy, a significant obstacle remains: sample efficiency, i.e., how efficiently a method finds a promising molecular modification with a limited number of molecular property evaluations. Evaluating the properties of a chemical compound is a time-consuming and costly process, and many existing methods in the literature require numerous trials, making them less practical for accelerated discovery. Consequently, there is a critical need for a methodology that can generate target chemical compounds in a more sample-efficient manner.

In this study, we introduce Conditional Latent Space Molecular Scaffold Optimization (CLaSMO) method, a framework designed to address two critical challenges in chemical compound discovery: real-world applicability and sample efficiency. CLaSMO combines a Conditional Variational Autoencoder (CVAE) (Higgins et al., 2016) with Latent Space Bayesian Optimization (LSBO) (Gómez-Bombarelli et al., 2018) to strategically modify input molecules and optimize their chemical properties.

In drug discovery, a common strategy for generating synthesizable molecules is to work with molecular scaffolds—key substructures that serve as the foundation for chemical modifications and drug design (Bemis & Murcko, 1996). Building on this approach, our framework integrates small substructures into these scaffolds to improve key molecular properties. For modifications to be both effective and synthesizable, it's crucial that the generative model understands how new substructures bond with the scaffold and enhance its properties. To achieve this, we condition substructure generation on the scaffold's chemical features using a CVAE. Our novel data preparation and training strategy enables the CVAE to generate substructures that align with specific atomic environments. We further optimize this process using LSBO, which efficiently explores both the latent space of the CVAE and the scaffold's chemical features. This allows CLaSMO to effectively select regions for modification and generate substructures that are chemically meaningful. By tracking molecular similarity between the initial scaffold and the optimized molecule, CLaSMO is designed to operate in low-budget scenarios and costly settings, such as wet-lab experiments, where high sample efficiency is crucial. It ensures that modifications are both realistic and applicable to real-world contexts. Ultimately, this approach accelerates the discovery of novel, synthesizable compounds with improved properties, as illustrated in Figure 1.

We evaluate our approach in several scenarios: optimizing the Quantitative Estimate of Drug-likeness (QED) in Section 5.2, where we also compare the performance of CLaSMO with other methodologies, and optimizing docking simulation scores in Section 5.3. QED assesses how likely a compound is to become a viable drug, while docking scores measure how well a molecule binds to a protein target using a simulator (Schrödinger, 2023). Furthermore, we benchmark our methodology against various competitor approaches across 20 additional molecular optimization tasks in Section 5.4, bringing the total number of optimization tasks evaluated for CLaSMO to 22. Our experiments demonstrate that CLaSMO efficiently improves both QED, docking

simulation scores, and many other benchmark optimization tasks in a sample-efficient manner, highlighting its effectiveness. Contributions of this study can be listed as follows:

1. We propose CLaSMO, a pioneering CVAE and LSBO-based molecule modification algorithm for molecular design. We use a novel data preparation strategy that enables CVAE to learn how substructures bond with target molecules, providing tailored generations.

2. We show that CLaSMO improves target properties with sample-efficiency while keeping the optimized molecules structurally similar to the input scaffolds, increasing the likelihood of identifying synthesizable compounds with desirable properties.

3. We demonstrate that CLaSMO achieves state-of-the-art results using a significantly smaller model and training dataset than its competitors, highlighting its efficiency.

4. We open source a web-application, https://clasmo.streamlit.app/, that enables interactive optimization of input molecules via CLaSMO, which allows chemical experts to decide the region to modify in input molecule, enabling Human-in-the-Loop optimization settings.

## 2 Related Works

Molecular design strategies can broadly be divided into two categories: from-scratch-generation of molecules and modification-based approaches. Both categories have made significant strides in recent years, yet they also face unique challenges, particularly regarding real-world applicability and sample efficiency.

From-scratch-generation approaches focus on creating entirely new molecules by conducting a search to optimize the target property. A seminal work by Gómez-Bombarelli et al. (2018) introduced latent space optimization-based methodology, using a VAE to generate novel compounds by navigating the latent space of molecular representations. LSBO builds on this by efficiently reducing the number of expensive black-box evaluations required for molecular optimization, enabling the discovery of compounds with desirable properties in a continuous latent space. Since then, numerous studies have further refined and expanded the LSBO framework, focusing on method development and practical applications (Grosnit et al., 2021; Tripp et al., 2020; Maus et al., 2022; Griffiths & Hernández-Lobato, 2020; Boyar & Takeuchi, 2024; Boyar et al., 2024). However, like many other generation-from-scratch methods, such methodologies struggle with real-world applicability—i.e., the difficulty of synthesizing the generated molecules in real-world settings (Lim et al., 2020). Other generation methods, such as genetic algorithms (Jensen, 2019), Grammar VAE (Kusner et al., 2017), and Junction Tree (JT) VAE (Jin et al., 2018), aim to improve the chemical validity of generated structures. Recent advancements like GP-MOLFORMER (Ross et al., 2024) utilizes large language model-like approaches for molecular design, but the real-world applicability challenge remains a major limitation across these methodologies.

Modification-based approaches, on the other hand, focus on adding substructures to existing molecules or scaffolds, often leading to more synthesizable and interpretable designs. Methods like Scaffold-GGM (Lim et al., 2020) employ a graph generative model to modify molecular scaffolds, thus improving properties with a higher chance of obtaining synthesizable molecules. Weller & Rohs (2024) introduces DrugHIVE, a deep hierarchical variational autoencoder that leverages scaffold modification to generate novel molecular compounds. There are many other scaffold-based optimization methodologies (Li et al., 2019; Langevin et al., 2020), not limited to generative modeling (Schreiber, 2000; Welsch et al., 2010; Miao et al., 2011). Techniques like Bradshaw et al. (2019)'s model ensure chemical validity in molecular modifications, and Gottipati et al. (2020)'s PGFS model uses reinforcement learning to guide additions to the base molecule. These approaches aim to avoid the low real-world viability problem faced by generation-from-scratch methods, as they build upon known molecular scaffolds, however, they lack advanced optimization methodologies that take sample efficiency into account.

CLaSMO combines the strengths of both categories, leveraging LSBO in the latent space of a CVAE for improved sample efficiency while focusing on scaffold-based modifications. Unlike current LSBO-based molecular design methodologies in the literature, CLaSMO does not generate molecules from scratch but instead

optimizes molecular properties by adding substructures to existing scaffolds. This approach mitigates the real-world applicability problem, and increases the chance of obtaining molecules that are both effective and practical for real-world synthesis. In order to evaluate the sample-efficiency of CLaSMO and benchmark it against other methodologies, we referred to the sample-efficiency benchmark proposed by Gao et al. (2022), which introduces various challenging molecular property optimization tasks for molecular design using various oracle functions, and benchmarks the performance of numerous methodologies from the literature. Among these, several approaches can be applied in a scaffold optimization setting. For example, genetic algorithm-based approaches such as Graph-GA (Jensen, 2019), which uses a graph-based genetic algorithm and generative model, Smiles-GA (Brown et al., 2019), which employs string-based SMILES (Weininger, 1988) representations of molecules, and Stoned (Nigam et al., 2021), which leverages SELFIES (Krenn et al., 2020), an alternative string-based molecular representation, are noteworthy examples for genetic algorithm based approaches. Additionally, reinforcement learning techniques like MolDQN (Zhou et al., 2019b) have also garnered significant attention for their effectiveness in molecular design, which is also applicable in scaffold optimization setting. Genetic algorithms, in particular, have been recognized for their robust performance in molecular design tasks, as discussed in Tripp & Hernández-Lobato (2023), where benchmarking against such methods is strongly recommended due to their continued competitiveness against more advanced techniques. We adopt the experimental setting proposed by Gao et al. (2022) in Section 5.4 and demonstrate that CLaSMO achieves superior sample efficiency and optimization performance across various molecular property optimization tasks. By incorporating a targeted scaffold-modification strategy, CLaSMO offers a balanced and efficient solution to molecular design.

## 3 Preliminaries and Problem Setup

In this section, we provide preliminary knowledge on CVAEs, LSBO, and scaffolds. We then discuss the challenges of property optimization and scaffold modifications.

### 3.1 Conditional Variational Autoencoders (CVAEs)

A VAE (Kingma & Welling, 2014) consists of an encoder $f_\phi^{\text{enc}} : \mathcal{X} \to \mathcal{Z}$ and a decoder $f_\theta^{\text{dec}} : \mathcal{Z} \to \mathcal{X}$, where $\mathcal{X}$ represents the input space and $\mathcal{Z}$ the latent space. CVAEs extend the framework of VAEs by incorporating additional condition vector $\boldsymbol{c}$ into the latent variable model, facilitating the controlled generation of new instances. In the CVAE architecture, the encoder $q_\phi(\boldsymbol{z}|\boldsymbol{x},\boldsymbol{c})$ maps an input $\boldsymbol{x}$ and a condition $\boldsymbol{c}$ to a latent representation $\boldsymbol{z}$. Simultaneously, the decoder $p_\theta(\boldsymbol{x}|\boldsymbol{z},\boldsymbol{c})$ reconstructs $\boldsymbol{x}$ using both $\boldsymbol{z}$ and $\boldsymbol{c}$. The training of CVAEs is formulated as the minimization of the conditional variational lower bound:

$$\mathcal{L}(\theta, \phi; \boldsymbol{x}, \boldsymbol{c}) = -\mathbb{E}_{q_\phi(\boldsymbol{z}|\boldsymbol{x},\boldsymbol{c})} \left[ \log p_\theta(\boldsymbol{x}|\boldsymbol{z},\boldsymbol{c}) \right] + \text{KL} \left( q_\phi(\boldsymbol{z}|\boldsymbol{x},\boldsymbol{c}) \, \| \, p(\boldsymbol{z}) \right), \tag{1}$$

where KL denotes the Kullback-Leibler divergence. In this model, the prior distribution $p(\boldsymbol{z})$ over the latent variables is typically assumed to be a standard normal distribution, $\mathcal{N}(0, I)$. This assumption simplifies the learning process by standardizing the latent space, ensuring that the encoder learns a distribution that closely aligns with a prior distribution, thus enhancing the generative capability of the decoder conditioned on specific contexts.

### 3.2 Latent Space Bayesian Optimization (LSBO)

In BO, we start with numerous *unlabeled* instances $\{\boldsymbol{x}_i\}_{i \in [\mathcal{U}]}$ and a smaller set of *labeled* instances $(\boldsymbol{x}_i, y_i)_{i \in [\mathcal{L}]}$, where an input $\boldsymbol{x}_i \in \mathcal{X}$ represents a chemical compound, and a label $y_i \in \mathcal{Y} \subseteq \mathcal{R}$ indicates its properties such as docking scores. BO seeks to optimize a costly black-box function $f^{\text{BB}} : \mathcal{X} \to \mathcal{Y}$, which corresponds to obtaining the physical properties of chemical compounds through experiments or time-consuming simulations in the context of molecular design problems. The goal is to maximize $f^{\text{BB}}$ with minimal evaluations, using typically a Gaussian Process (GP) surrogate, trained using the labeled instances $\mathcal{L}$, to predict the function over $\mathcal{X}$. BO uses the surrogate to select an input $\boldsymbol{x}$ that may yield values surpassing the current maximum $\max_{i \in \mathcal{L}} y_i$. However, building a GP surrogate in high-dimensional spaces like chemical compounds is challenging. LSBO tackles this by employing a VAE/CVAE trained on the unlabelled instances $\mathcal{U}$ to

Figure 2: Examples of scaffold extraction from whole molecules. In both the upper and lower rows, side chains are removed, leaving the core structure of the molecule. These resulting scaffolds act as starting points for novel chemical design.

reduce dimensionality, encoding instance in $\mathcal{X}$ to lower dimensional latent space $\mathcal{Z}$. This simplifies surrogate modeling and optimization because $\mathcal{Z}$ has lower-dimension than $\mathcal{X}$. During LSBO iterations, the acquisition function applied to the GP's predictions selects new points in $\mathcal{Z}$ to evaluate. The chosen latent variable $\boldsymbol{z}_{i'}$ is decoded into a new input $\boldsymbol{x}_{i'} = f_\theta^{\mathrm{dec}}(\boldsymbol{z}_{i'})$. This new instance is evaluated by $f^{\mathrm{BB}}$, and the results update $\mathcal{L}$ and refine the GP model. This cycle repeats until optimal results are achieved or resources are exhausted. In contexts like molecular design, LSBO aims to discover chemical compounds with optimal properties by efficiently navigating the reduced latent space.

### 3.3 Scaffolds

Scaffolds (Bemis & Murcko, 1996) are the stable core structures within molecules that serve as the framework for chemical modifications in drug design. Scaffolds retain the essential biological activity of the molecule. They play a crucial role in molecular design by providing a foundation for chemical modifications aimed at optimizing properties like QED. Researchers often use scaffolds to systematically explore chemical variations (Schreiber, 2000; Welsch et al., 2010), which can lead to the discovery of new compounds with improved properties.

In this study, we follow the scaffold extraction method from Bemis & Murcko (1996), where non-essential components like side chains are removed, leaving the core structure. This extracted scaffold serves as a starting point for further modifications, allowing efficient exploration of chemical space. By focusing on scaffolds, molecular design becomes more streamlined, increasing the likelihood of identifying novel, synthesizable compounds with the desired biological activity. Examples of whole molecule and scaffold pairs are provided in Fig. 2.

### 3.4 Problem Definition

We denote the scaffold, which is the base of the modification, as $\boldsymbol{S}$, and the modified molecule as $\boldsymbol{S}'$. Our goal is to efficiently find the modification that maximizes the molecular property P which is evaluated by $f^{\mathrm{BB}}(\boldsymbol{S}')$, while keeping the difference between $\boldsymbol{S}$ and $\boldsymbol{S}'$ small. Directly iterating over all possible $\boldsymbol{S}'$ to find the best modification that maximizes $f^{\mathrm{BB}}(\boldsymbol{S}')$ is impractical, as it involves high complexity and costly evaluations of $f^{\mathrm{BB}}$.

The primary challenge in optimizing molecular scaffolds lies in i) determining the optimal bonding point on the base scaffold $\boldsymbol{S}$, and ii) selecting the appropriate substructures added to the bonding point to ensure meaningful improvements in the desired property $\mathcal{P}$. A molecular scaffold $\boldsymbol{S}$, composed of several atoms $p_1, p_2, \ldots, p_k$, may have atoms with the remaining capacity to form additional chemical bonds. These atoms serve as potential candidates for bonding with newly generated substructures. Therefore, the task involves not only selecting the right substructure but also identifying the most suitable bonding point $p_i$ to optimize scaffold properties. This adds complexity, as the need for precise modifications must be balanced with the challenges of high-dimensional search spaces and evaluation costs. Consequently, a more efficient approach is required to explore scaffold modifications effectively while minimizing the number of evaluations.

To address this, the problem can be reframed as an optimization task in a reduced latent space $\mathcal{Z}$, obtained through a CVAE. In this space, each point $\boldsymbol{z} \in \mathcal{Z}$ corresponds to a potential substructure that can be integrated into the scaffold. By encoding the molecular substructures into this lower-dimensional space $\mathcal{Z} \in \mathbb{R}^d$, the search becomes more tractable. The objective is to find the optimal latent representation $\boldsymbol{z^*}$ conditioned on the optimal bonding point $p_i^*$ that, together, maximize the desired property $\mathcal{P}$ when the generated substructure $\boldsymbol{s'} \leftarrow f^{\text{dec}}(\boldsymbol{z})$ is added to the scaffold. Let us denote this modification as $\boldsymbol{S}' \leftarrow g(\boldsymbol{S} \oplus \boldsymbol{s'}, p_i)$, where the function $g()$ adds substructure $\boldsymbol{s}$ to the scaffold $\boldsymbol{S}$ at $p_i$. The optimization problem is then formulated as:

$$\boldsymbol{z^*}, p_i^* = \arg \max_{\boldsymbol{z} \in \mathcal{Z}, p_i \in B(\mathbf{S})} f^{\text{BB}}(\boldsymbol{S}') = \arg \max_{\boldsymbol{z} \in \mathcal{Z}, p_i \in B(\mathbf{S})} f^{\text{BB}} g(\boldsymbol{S} \oplus f^{\text{dec}}(\boldsymbol{z}), p_i), \tag{2}$$

where $B(\boldsymbol{S})$ is the set of possible bonding points on the scaffold $\boldsymbol{S}$. In Section 4.2, we demonstrate that atomic features at $p_i$ are used as condition vectors to generate new substructures, enabling targeted substructure generation for atom $p_i$.

### 3.4.1 Controlling Molecular Similarity

Current modification-based methods often fail to account for how changes impact molecular similarity between the original scaffold $\boldsymbol{S}$ and the updated scaffold $\boldsymbol{S}'$, or any structure in general. Adding substructures typically increases molecular weight, which can hinder real-world applicability, especially when exceeding 500 Daltons, as indicated by Lipinski's Rule of Five (Lipinski et al., 2001). Higher molecular weight compounds are more difficult to synthesize, making them less suitable for molecular design. Additionally, it is sometimes necessary to ensure that modifications result in only minor adjustments to avoid drastic changes.

Thus, a key challenge is to guide the optimization process by considering molecular similarity, ensuring that the modified molecules remain structurally close to the original scaffold. Such a framework can increase the likelihood of obtaining synthesizable compounds by limiting divergence from known molecules. In Section 5, we show that our method effectively solves the optimization problem in Eq. (2) while ensuring molecular similarity between $\boldsymbol{S}$ and $\boldsymbol{S}'$.

## 4 Proposed Method

Our proposed CLaSMO framework comprises two key components: the CVAE and the LSBO algorithm. However, an essential first step in our approach is the data preparation required to train the CVAE. In this section, we will begin by outlining the data preparation process, followed by an explanation of the CVAE and the CLaSMO methodology.

### 4.1 Data Preparation

Our proposed method requires a uniquely tailored dataset because no existing dataset in the literature fully meets the specific needs of our approach. To create this dataset, we developed a BRICS (*Breaking Retrosynthetically Interesting Chemical Substructures*) (Degen et al., 2008) based approach. BRICS is an algorithm designed to decompose organic molecules into smaller, synthetically feasible substructures by identifying breaking points within a molecule's structure based on chemical retrosynthetic rules. These breaking points, also known as division points, are the connections between subgraphs within the molecule that BRICS identifies. The algorithm systematically breaks down a molecule $\boldsymbol{M}$ into $k$ substructures, Fig. 3 demonstrates this procedure. BRICS is particularly well-suited for this task because it ensures that the generated substructures are synthetically feasible and chemically valid, making it ideal for scaffold-based molecular optimization. Unlike other decomposition methods, BRICS explicitly adheres to retrosynthetic rules, ensuring compatibility with real-world chemical synthesis.

The division points found by BRICS are of particular importance because they serve as both the points where the molecule is split into substructures and the potential bonding sites where these substructures can be reattached. Therefore, they provide us a valuable information about these substructures in terms of what kind of bonds they can form. This dual role makes the division points a crucial piece of information

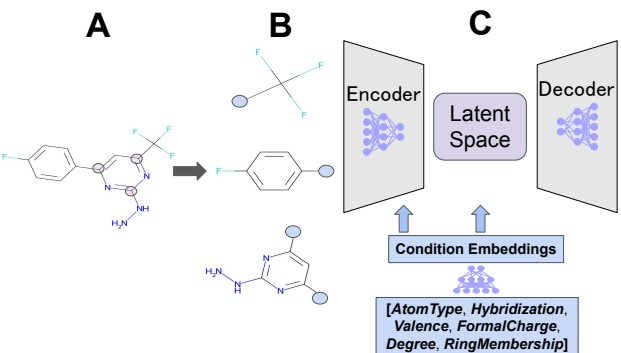

Figure 3: Illustration of the BRICS algorithm and its integration with the CVAE. The molecule in **A** is decomposed into substructures using the BRICS algorithm (**B**), with specific breaking points highlighted. Atomic environment features are extracted from these points to provide critical information about the bonding environment. Embeddings of these features are used as condition vectors, which are concatenated both at the encoder input along with the substructures and at the latent space of the CVAE (**C**), guiding the generation of substructures that are compatible with the scaffold.

for our generative model. By preserving the properties and features of atoms at these division points, we capture the chemical context around the atom that dictates how substructures can bond with other parts of a molecule. The atomic features we consider are *atom type, hybridization, valence, formal charge, degree, and ring membership*. Detailed explanations of these features are provided in the Appendix A.1. These features, when used as condition vectors in our CVAE, are essential for guiding the generation of substructures that can successfully bond with the scaffold, as they provide a detailed understanding of the chemical environment at each division point, such as their atom types and if the selected atom has the capacity for forming additional bonds. Therefore, our dataset for CVAE training includes the substructures obtained via the BRICS algorithm and six different atomic features of the atom at their breaking points. These substructures are represented using a string-based representation method referred to as SELFIES Krenn et al. (2020). The illustration of the BRICS algorithm, atomic feature extraction points, and their integration with CVAE is provided in Fig. 3.

### 4.2 Substructure CVAE

Our CVAE consists of an encoder, parameterized by $f_\phi^{\text{enc}}$, and a decoder, parameterized by $f_\theta^{\text{dec}}$. The encoder takes the substructure $s$ and the associated condition vector $c$, and maps this input into a latent space, producing a latent representation $z$. The decoder then takes a point $z$ from this latent space, along with the condition vector $c$, and generates a substructure $s'$ that is conditioned on the given atomic properties. The loss function of the proposed CVAE is defined as:

$$\mathcal{L}(\phi, \theta; s, c) = \|s - \mathbb{E}_{q_\phi(z|s,c)}[p_\theta(s|z,c)]\|^2 + \beta \cdot D_{\text{KL}}(q_\phi(z|s,c)\|p(z)), \tag{3}$$

where $q_\phi(z|s,c)$ is the encoder, $p_\theta(s|z,c)$ is the decoder, and $\beta$ balances the reconstruction and KL divergence terms (Higgins et al., 2016). The condition vectors allow the CVAE to learn how different substructures interact with specific atomic environments, building a deeper understanding of their bonding behavior. Figure 3B demonstrates the input substructures that CVAE uses in its training, and Fig. 3C visualizes the CVAE architecture along with the condition vectors.

After training, if during the generation phase, we provide the CVAE with a condition vector that specifies the atomic environment of the target scaffold, the decoder, using a latent vector along with the condition vector, generates substructures that are tailored to bond effectively with the scaffold. This conditioning mechanism ensures that the generated substructures are not random but specifically designed to fit the scaffold's atomic environment, improving the efficiency of scaffold modifications and the likelihood of successful bonding. In the proposed CLaSMO method, we jointly optimize the bonding position $p_i$ and the latent vector $z$,

conditioned on the atomic properties $\boldsymbol{c}$ of the bonding site in order to optimize both the bonding position and the substructure added to the position.

### 4.2.1 Condition Vector Embeddings

In Section 4.1, we detailed our data preparation process and the extraction of six atomic environmental features, resulting in a 6-dimensional condition vector. As we will explain in Section 5.1, the relatively simple nature of the substructures used in CVAE training allows for a much lower latent dimension than the actual conditional vector dimension of 6. To align with this simplicity and ensure efficient representation, we employed an Autoencoder model to generate $d$-dimensional embeddings of the atomic features. These embeddings preserve the key characteristics within the atomic environments and learn the relationship among distinct atomic features while reducing dimensionality, which helps lower computational complexity and improves the model's ability to generalize. After its training, the encoder of the Autoencoder, $f_{\mathrm{CondEmd}}^{\mathrm{enc}}$, is used to provide the final condition vector $\boldsymbol{c}$ to be inputted to CVAE by encoding the six-dimensional atomic environmental feature vector. This process is visualized in Fig. 3C.

### 4.3 CLaSMO

In CLaSMO, upon training of the CVAE, we perform a targeted search in the latent space $\mathcal{Z} \in \mathbb{R}^d$ of the CVAE, where decodings from each point $\boldsymbol{z}$ represent a potential substructure to be added to the scaffold. The challenge is to modify the scaffold $\boldsymbol{S}$ by selecting a substructure and integrating it at an appropriate bonding point $p_i$, optimizing the desired molecular property $\mathcal{P}$. Therefore, the search space $\Omega$ we consider in our optimization tasks is defined as the Cartesian product of the latent space and the set of possible bonding points on the scaffold:

$$\Omega = \mathbb{R}^d \times \{p_1, p_2, \ldots, p_n\}.$$

Within this search space, the goal is to modify the scaffold $\boldsymbol{S}$ to maximize the BB function $f^{\mathrm{BB}}(\boldsymbol{S'})$.

However, although our main goal is optimizing the target property, we also aim to keep the similarity between $\boldsymbol{S}$ and $\boldsymbol{S'}$ at a certain level. Therefore, in CLaSMO, we apply a similarity constraint using the *Dice Similarity* (Dice, 1945) metric to compare the input scaffold $\boldsymbol{S}$ with the modified molecule $\boldsymbol{S'}$ before evaluating the BB function. In order to measure the similarity between $\boldsymbol{S}$ and $\boldsymbol{S'}$, we use Morgan Fingerprints (Morgan, 1965; Rogers & Hahn, 2010). The Morgan fingerprint of a molecule is represented as a binary vector, where each bit indicates the presence or absence of a specific substructure within the molecule, making them effective and popular for comparing molecular similarities. Using these, the similarity between $\boldsymbol{S}$ and $\boldsymbol{S'}$ is measured by computing the Dice Similarity of their Morgan fingerprint vectors[1], defined as:

$$\mathrm{DICE}(\boldsymbol{S}, \boldsymbol{S'}) = \frac{2|\boldsymbol{M_S} \cap \boldsymbol{M'_S}|}{|\boldsymbol{M_S}| + |\boldsymbol{M'_S}|}$$

where $\boldsymbol{M_S}$ and $\boldsymbol{M'_S}$ are the Morgan fingerprint vectors of $\boldsymbol{S}$ and $\boldsymbol{S'}$, respectively. Dice Similarity, ranging from 0 (no overlap) to 1 (identical), helps us track structural changes during optimization.

The final optimization objective, incorporating both the search for optimal substructures and the similarity constraint, is given by:

$$\boldsymbol{z^*}, p_i^* = \arg \max_{(\boldsymbol{z}, p_i) \in \Omega} f^{\mathrm{BB}}(\boldsymbol{S'}) \qquad (4)$$

$$\text{subject to:} \quad \mathrm{DICE}(\boldsymbol{S}, \boldsymbol{S'}) \geq \tau,$$

where $\boldsymbol{S'} \leftarrow g(\boldsymbol{S} \oplus f^{\mathrm{dec}}(\boldsymbol{s'}|\boldsymbol{z^*}, \boldsymbol{c}))$. By this approach, CLaSMO efficiently navigates the latent space, optimizing molecular properties while allowing modifications only if $\mathrm{DICE}(\boldsymbol{S}, \boldsymbol{S'}) \geq \tau$, effectively controlling the degree of divergence from the input scaffold.

For the joint optimization of substructures and bonding positions, we consider a GP model: $(\boldsymbol{z}, p) \mapsto y'_{\Delta}$, where $\boldsymbol{z}$ is the latent vector representing a substructure, $p$ is the bonding position of the modification, and

---

[1]Bajusz et al. (2015) discusses that the dice similarity is one of the best metrics to assess similarity between molecules using Morgan fingerprints.

$y'_\Delta := y - y'$ represents the improvement in the property, with $y$ and $y'$ indicating the properties of molecules $S$ and $S'$, respectively. To prepare the training dataset $\mathcal{D}$ for the GP, we conducted a random sampling of substructures from random latent vectors $z$, each paired with randomly selected bonding regions $p$ on the scaffold $S$, and evaluated the $y'_\Delta$ obtained from these additions (e.g., in the case of QED optimization, QED score differences between $S$ and $S'$ are calculated). These triplets of $(z, p, y'_\Delta)$ are used in GP training. This setup allows the GP model to learn the relationship between the latent space representations, atoms in the input scaffold, and the resulting property changes, guiding the optimization process toward regions of the latent space that are more likely to yield beneficial modifications to the scaffold.

Among many possible choices, we employ the Upper Confidence Bound (UCB) acquisition function to guide the optimization process. To ensure LSBO samples from regions that meet the similarity constraint, we introduced a penalization mechanism. Specifically, we assign a negative improvement value $y'_\Delta$ when the condition $\text{DICE}(S, S') \geq \tau$ is not satisfied after sampling from $f^{\text{dec}}([z^*, c])$. Additionally, we also apply a penalty when a substructure cannot bond with the target molecule. We outline our approach in Algorithm 1, and provide details about the selection of hyperparameters within our framework in the Appendix A.4.

### 4.3.1 Kernel Design of CLaSMO

The problem we try to solve requires simultaneous optimization over continuous latent vectors and discrete bonding points. To handle this complexity, we use a GP model with a covariance function $k$ that accommodates the mixed input space of continuous and discrete variables. We define separate kernels for these inputs:

$$k_{\text{cont}}(z, z') = \exp\left(-\frac{1}{2\ell^2}\|z - z'\|^2\right), \quad k_{\text{cat}}(p_i, p_j) = \exp\left(-\frac{\delta_{p_i, p_j}}{\ell}\right),$$

where $k_{\text{cont}}(z, z')$ is an RBF kernel, and $k_{\text{cat}}(p_i, p_j)$ measures the similarity between atoms in the molecule that are ready for additional bond via Kronecker delta function, measuring equality of atom positions, and $\ell$ is the lengthscale parameter. The combined kernel used in CLaSMO is then expressed as:

$$k_{\text{CLaSMO}} = k((z, p_i), (z', p_j)) = k_{\text{cont}}(z, z') \cdot k_{\text{cat}}(p_i, p_j).$$

## 5 Experiments

In this section, we first provide details about the training of our CVAE model in §5.1. Next, we discuss the results from two different experimental settings, which are Quantitative Estimate of Drug-likeness (QED) property optimization and docking simulation score property optimization in Sections 5.2 and 5.3. Our focus on QED property optimization stems from the fact that it is one of the most popular optimization tasks in molecular design studies, which enables us to benchmark our results with many other studies. The docking simulation score property, on the other hand, provides a close-to-real-life setting as we calculate docking scores using an accurate, computation-heavy simulator tool, which enables us to evaluate the performance of CLaSMO in such a setting. We also use these experiments to analyze the impact of keeping the similarity between input and optimized scaffold at different threshold values on optimization performance. On the other hand, in order to provide a more extensive benchmark of CLaSMO against other methodologies, we refer to the sample-efficiency benchmark setting for molecular property optimization proposed by Gao et al. (2022) in Section 5.4, and run CLaSMO in 20 additional property optimization settings with varying complexities and compare its performance with competitor models, showcasing the sample-efficiency of CLaSMO against other popular approaches in the field.

### 5.1 CVAE Training

For our data preparation, we used the QM9 dataset (Ruddigkeit et al., 2012; Ramakrishnan et al., 2014), which contains 130,000 small molecules. QM9 was chosen for its simplicity and suitability for our task, as it consists of smaller molecules compared to other well-known datasets like ZINC (Irwin & Shoichet, 2005; Irwin et al., 2020). Using our data preparation strategy, we extracted 18,706 unique substructure pairs along with their atomic environment features to train our CVAE. Most of the substructures obtained from QM9

---

**Algorithm 1** CLaSMO

---

1: **Input:** GP training data $\mathcal{D}$, Trained CVAE, Trained Autoencoder encoder $f_{\text{CondEmd}}^{\text{enc}}$ for condition embeddings, Input molecules $\boldsymbol{M}$, Optimization budget per molecule $K$, Similarity threshold $\tau$, Penalization terms $\lambda_1, \lambda_2$
2: Fit GP using $\mathcal{D}$
3: **for** $\boldsymbol{i} = 1$ to $\boldsymbol{M}$ **do**
4:     Pick molecule $\boldsymbol{m}_i$, obtain its scaffold $\boldsymbol{S}_i \leftarrow \text{Scaffold}(\boldsymbol{m}_i)$,
5:     Evaluate target property value $y \leftarrow f^{\text{BB}}(\boldsymbol{m}_i)$
6:     **for** $j = 1$ to $K$ **do**
7:         Identify available atoms in the scaffolds for bonding, $p_j \in B(\boldsymbol{S})^2$
8:         Find $\boldsymbol{z^*}, p_i^* = \arg\max_{(\boldsymbol{z}_i, p_i) \in \Omega} f^{\text{BB}}(S')$
9:         Create condition vector $\boldsymbol{c^*}$ for atom $p_i^*$ in molecule $\boldsymbol{S}_i$
10:        Obtain condition embeddings $\boldsymbol{c} \leftarrow f_{\text{CondEmd}}^{\text{enc}}(\boldsymbol{c^*})$
11:        Generate substructure $\boldsymbol{s^*} \leftarrow f^{\text{dec}}([\boldsymbol{z^*}, \boldsymbol{c}])$
12:        Add substructure $\boldsymbol{s^*}$ to molecule $\boldsymbol{S}_i$ at region $p_i^*$ to obtain $\boldsymbol{S}_i'$
13:        **if** $\boldsymbol{S}_i \neq \boldsymbol{S}_i'$ **then**
14:            **if** $\text{DICE}(\boldsymbol{S}_i, \boldsymbol{S}_i') > \tau$ **then**
15:                Evaluate new property: $y' = f^{\text{BB}}(\boldsymbol{S}_i')$
16:                Compute improvement $y_\Delta' = (y - y')$
17:                **if** $y_\Delta' > 0$ **then**
18:                    Update $\boldsymbol{S}_i \leftarrow \boldsymbol{S}_i'$
19:                **end if**
20:            **else**
21:                Set $y_\Delta'$ to penalization term $\lambda_1$
22:            **end if**
23:        **else**
24:            Set $y_\Delta'$ to penalization term $\lambda_2$
25:        **end if**
26:        Update $\mathcal{D} \leftarrow \mathcal{D} \cup \{[\boldsymbol{z^*}, p^*], y_\Delta'\}$
27:        Update GP with $\mathcal{D}$
28:     **end for**
29: **end for**

---

through our method are under 100 Daltons, making them ideal for training a model focused on generating small substructures to modify the input scaffold. Since our goal is to optimize the scaffold while maintaining a high degree of similarity in the modifications, using such a dataset allows CVAE to learn to generate small substructures, which is crucial for achieving our goals. Of the 18,706 instances, 80% were used for model training, with the remaining data allocated for testing and validation. We represented the molecules using SELFIES (Krenn et al., 2020), a string-based molecular representation, in the form of one-hot encoding matrices. Our CVAE architecture consists of three fully connected layers in the encoder and three GRU layers in the decoder.

Given the simplicity of the dataset, we determined that a 2-dimensional latent space was sufficient to achieve over 99% reconstruction accuracy on the test set. This lower-dimensional space also enhances the performance of BO, which typically excels in smaller-dimensional spaces. Additionally, to maintain this simplicity, as explained in Section 4.2.1, we generated a 2-dimensional embedding for the condition vectors using an Autoencoder model, which is trained with fully connected layers in both the encoder and decoder, achieved 93% reconstruction accuracy for the condition vectors. Prior to CVAE training, we obtained the condition vector embeddings via the Autoencoder model, and used them during CVAE training. As a result, the CVAE was trained with both a 2-dimensional latent space for the substructures and 2-dimensional condition

---

[2]This identification is performed by finding atoms within the molecule that have additional capacity to form bonds. For example, a carbon atom can form up to four bonds, and if it has only three bonds within the molecule, it will be identified as available for bonding.

vectors, effectively balancing simplicity and optimization performance. Further discussion on this design choice is provided in Appendix A.3.

## 5.2 Quantitative Estimate of Drug-likeness (QED) Optimization

In this section, we present the results of our QED optimization experiments, where we used RDKit (Landrum, 2010) to calculate QED values. The QED metric, defined between 0 and 1, inherently limits the range of potential improvements, making optimization within this closed range particularly challenging. To further complicate this task, we imposed similarity constraints to ensure the optimized molecules remain close to their original scaffolds, by running the CLaSMO algorithm with $\tau$ values $\tau \in [0, 0.25, 0.5, 0.6]$. For each threshold, CLaSMO was run for 100 iterations on 100 input scaffolds, where their whole molecules are sampled randomly from the ZINC250K dataset (Gómez-Bombarelli et al., 2018). The search domain for each latent dimension in $\mathcal{Z}$ is set to $[-6, 6]$. The GP model is trained using 100 training instances. Such an experimental setting is designed to demonstrate the optimization capabilities of CLaSMO at changing similarity thresholds, within the limited range of QED optimization tasks.

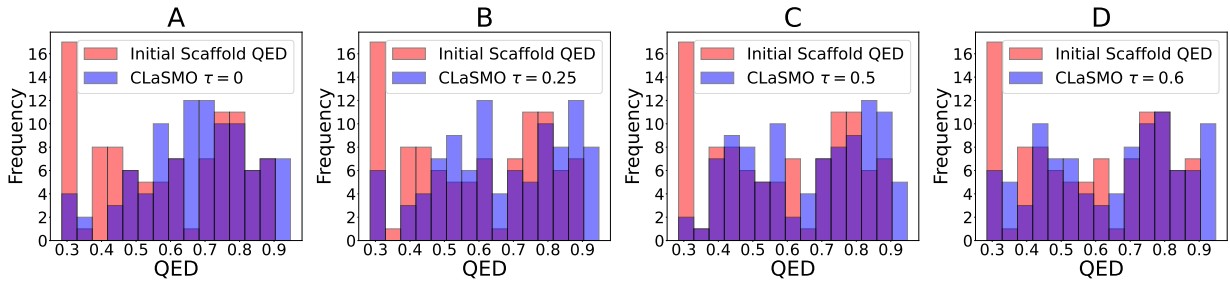

Figure 4: Distribution of QED values of input scaffolds and CLaSMO optimization results at various similarity thresholds, illustrating the shift towards higher QED values at each similarity level. (A) $\tau = 0$, showing the greatest improvement due to no similarity constraints. (B) $\tau = 0.25$, demonstrating high QED gains. (C) $\tau = 0.5$ and (D) $\tau = 0.6$ are still able to demonstrate significant improvements while enforcing a balance between optimization and similarity.

Initially, the average QED score of the input scaffold molecules was 0.5876, with a maximum QED of 0.8902. After optimization, CLaSMO with no similarity constraint ($\tau = 0$) achieved the highest QED score of 0.9480 and an average QED of 0.6778, representing a mean improvement rate of 21.43%[3] and a maximum improvement rate of 81.17%. As the similarity constraint was tightened, the performance remained competitive. With $\tau = 0.25$, the maximum QED score was 0.9464, with an average QED of 0.6715, leading to a mean improvement rate of 19.14%. Further increasing the similarity threshold to $\tau = 0.50$, the maximum QED score was 0.9437, and the average QED was 0.6602, yielding a mean improvement rate of 16.08%. For $\tau = 0.60$, the maximum QED score was 0.9402, and the average QED was 0.6434, resulting in a mean improvement rate of 11.70%. The results clearly demonstrate that CLaSMO consistently enhances the QED values across a range of similarity thresholds, effectively balancing the trade-off between maximizing QED and preserving structural similarity. Notably, even under more stringent similarity constraints, CLaSMO achieves significant improvements over the initial QED values, highlighting its robust optimization capabilities. This consistent performance across different thresholds underscores CLaSMO's versatility and effectiveness in optimizing QED, making it a powerful tool for scaffold-based molecular optimization tasks. Table 1 summarizes these findings. These results were achieved with only 100 iterations per input scaffold, demonstrating the sample efficiency of our approach. Additionally, Fig. 4 demonstrates two example sets of scaffold optimization results from the CLaSMO experiments, in which it can be observed that our algorithm finds a molecule with a QED score of 0.948 by very limited additions, keeping the similarity between the input and optimized molecule at the high level.

---

[3]Mean improvement rates calculated as the average of the relative improvements obtained by CLaSMO for each input scaffolds.

Table 1: Summary of QED optimization results: Max and mean QED values from input scaffolds and CLaSMO experiments with various $\tau$ levels, as well as mean and max improvement rates over QED values of input scaffold.

| Metric | Input | $\tau = 0$ | $\tau = 0.25$ | $\tau = 0.50$ | $\tau = 0.60$ |
|---|---|---|---|---|---|
| Max QED | 0.8902 | 0.9480 | 0.9464 | 0.9437 | 0.9402 |
| Mean QED | 0.5876 | 0.6778 | 0.6715 | 0.6602 | 0.6434 |
| Mean Imp. | - | 21.43% | 19.14% | 16.08% | 11.70% |
| Max Imp. | - | 81.17% | 114.57% | 74.74% | 72.41% |

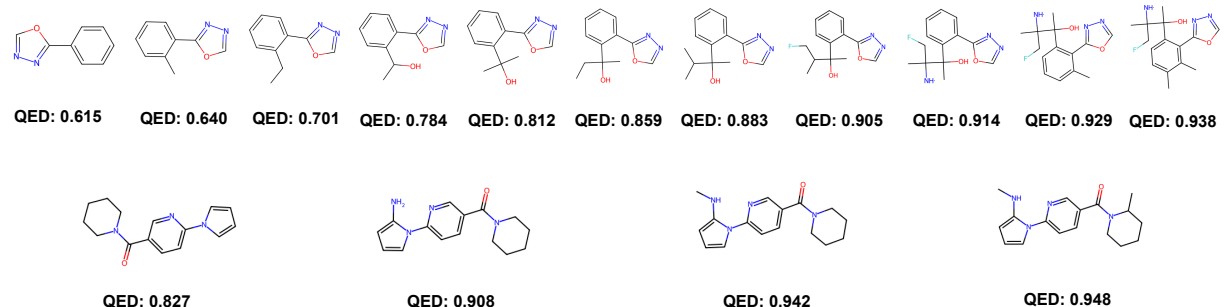

Figure 5: Examples of molecules obtained from different CLaSMO experiments in the QED setting. In both rows, the molecules on the left represent the input scaffolds. Each molecule to the right shows a step of substructure addition along with its resulting QED score. The results demonstrate QED improvements from 0.615 to 0.938 and from 0.827 to 0.948 in these examples.

### 5.2.1 Model Benchmark

Table 2 shows the top QED values achieved by various models, highlighting CLaSMO's performance. Our method reaches a top QED score of 0.948, matching state-of-the-art results from PGFS (Gottipati et al., 2020) and GP-MOLFORMER (Ross et al., 2024). Among these models, Scaffold-GGM, DrugHIVE, and PGFS are substructure-addition-based methodologies, while the others generate molecules from scratch. A key distinction is that CLaSMO's CVAE is trained only on substructures, not complete molecules, using a dataset of only 18,706 instances. The maximum QED value of the substructure among these instances is 0.53, with average QED value overall 0.427[4]. In contrast, competitor models are trained on significantly larger datasets and full molecules, often encountering molecules with high QED values during training. These results underscore CLaSMO's efficiency and lower computational cost, demonstrating the advantages of using carefully curated datasets and tailored methodologies for the target task.

Table 2: Comparison of top QED values achieved by various models with their dataset sizes.

| Model | Top QED | Dataset Size |
|---|---|---|
| JT-VAE (Jin et al., 2018) | 0.925 | 250,000 |
| Scaffold-GGM (Lim et al., 2020)[6] | 0.928 | 349,809 |
| DrugHIVE (Weller & Rohs, 2024) | 0.940 | 27 Million |
| LIMO (Eckmann et al., 2022) | 0.947 | 250,000 |
| PGFS (Gottipati et al., 2020) | **0.948** | - |
| GP-MOLFORMER (Ross et al., 2024) | **0.948** | 1.1 Billion |
| **CLaSMO (ours)** | **0.948** | **18,706** |

---

[4]Distribution of the QED values used in CVAE training is provided in the Appendix A.2.

### 5.3 Docking Simulation Score Optimization

In this section, we present our docking simulation score optimization results using the KAT1 protein as the target. KAT1 is an ion channel protein, and compounds with favorable docking scores are more likely to bind effectively and influence its function, which is crucial for developing therapeutic agents targeting ion channel-related conditions[5]. By optimizing docking scores, we aim to identify compounds with the potential to modulate ion channel function and develop novel therapeutic strategies. Unlike the relatively simple calculation of QED values, optimizing docking scores for KAT1 requires computationally intensive simulations. For these calculations, we used Schrödinger Software (Schrödinger, 2023), known for providing reliable and accurate results, with each simulation taking anywhere from a few minutes to several hours. Given the high computational cost of these simulations, CLaSMO's sample-efficient approach is particularly valuable, enabling effective optimization while minimizing the number of expensive evaluations.

Similar to the QED optimization setting, the search domain for each latent dimension in $\mathcal{Z}$ was set to $[-6, 6]$. To address the high cost of obtaining labeled data for docking scores, we leveraged the same training set used in the QED experiments, treating them as low-fidelity instances for training the GP model. This approach allows the surrogate model to learn the relationships between atoms and latent vectors, even though the labels originate from QED rather than docking scores. By reusing this shared data, we enhance the GP model's capacity to generalize and effectively guide the optimization process, reducing the dependency on extensive docking score evaluations. Given the significant time required for each docking score calculation, we limited the experiments to 10 compounds from the ZINC250K dataset and ran CLaSMO for 100 iterations per compound using two Dice similarity thresholds, $\tau \in [0.25, 0.50]$. This setup ensures that despite the computational expense, we can still obtain comprehensive and meaningful results for docking score optimization, and observe if CLaSMO is capable of optimizing docking scores under similarity constraints.

We present the distribution of docking scores obtained from both CLaSMO experiments in Fig. 6. The results clearly demonstrate that CLaSMO, at both similarity thresholds, achieves significant improvements in docking scores compared to the initial scaffolds. Figure 7 provides detailed metrics, including the maximum and average improvements across the models, as well as the initial best values. Notably, CLaSMO with a similarity threshold of $\tau = 0.25$ achieved improvements of up to 96.3% over the initial scaffold docking score, while CLaSMO with $\tau = 0.50$ achieved improvements of up to 75.1%, both indicating strong optimization performance despite the limited number of LSBO iterations, showcasing CLaSMO's sample-efficiency. In Fig. 8, we provide example molecules obtained from our experiment, where we observe that, even at the beginning stages of the optimization where $\mathrm{DICE}(\boldsymbol{S}, \boldsymbol{S'})$ still above 0.70, we observe substantial improvements.

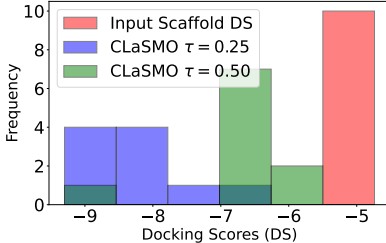

Figure 6: Distribution of docking scores for CLaSMO optimization experiments. Lower docking scores indicate better binding affinity, highlighting the efficacy of CLaSMO in enhancing docking performance across different similarity constraints.

| Metric | Input | $\tau = 0.25$ | $\tau = 0.50$ |
|--------|-------|---------------|---------------|
| Min DS | -5.051 | -9.30 | -8.57 |
| Mean DS | -4.934 | -8.32 | -6.62 |
| Mean Imp. | - | 70.1% | 35.4% |
| Max Imp. | - | 96.3% | 75.1% |

Figure 7: Summary of docking score (DS) optimization results: Max and mean DS values from input scaffolds and CLaSMO experiments with various $\tau$ levels, and mean and max improvement rates over DS values of input scaffold.

---

[5]Details of the KAT1 protein can be found at `https://pdbj.org/mine/summary/6v1x`.
[6]The results for Scaffold-GGM in this and the following experiments were obtained by running their released code.

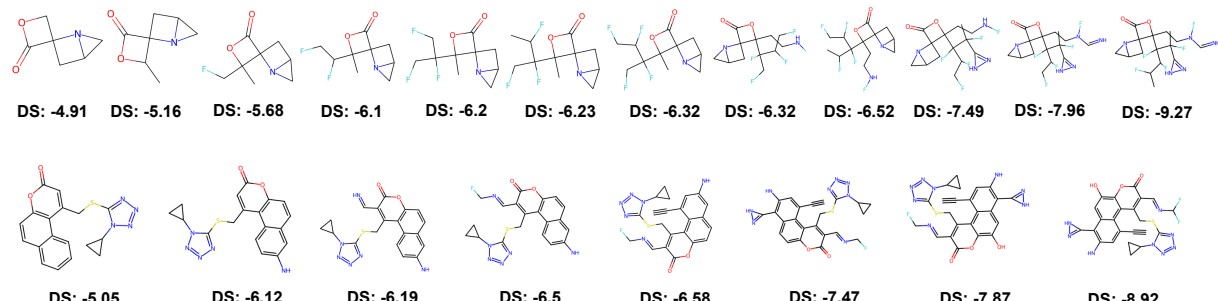

Figure 8: Examples of molecules obtained from CLaSMO $\tau = 0.25$ experiments from docking score optimization setting. In both rows, molecules in the left refer to the input scaffolds. Each molecule to the right shows one step of substructure addition with their resulting docking scores. Docking scores are significantly improved by adding small substructures to different regions of the input molecule.

## 5.4 Sample-Efficiency Benchmark

To comprehensively evaluate the sample-efficiency of our methodology, we applied CLaSMO to the sample-efficiency benchmark proposed by Gao et al. (2022), where we use 20 diverse molecular optimization tasks. These tasks span various objectives, including a broad range of pharmaceutically-relevant oracle functions, and challenging multi-property optimization (mpo) scenarios (Brown et al., 2019). Experiments were conducted with a fixed budget of 100 oracle evaluations per seed across 10 seeds, ensuring consistent comparisons. The choice of 100 oracle iterations (in contrast to the 10,000 iterations used in Gao et al. (2022)) reflects our focus on developing a methodology tailored for resource-constrained scenarios, which are more representative of real-world applications such as wet-lab experiments[7].

CLaSMO was benchmarked against five widely used baseline methods—Smiles-GA (Brown et al., 2019), Graph-GA (Jensen, 2019), Stoned (Nigam et al., 2021), MolDQN (Zhou et al., 2019a), and Scaffold-GGM (Lim et al., 2020)—using the same set of 100 scaffolds for all methods as starting molecules. No similarity constraints were applied in this experimental setting, and open-sourced implementations of all baseline methods were used with their pre-trained models.

Table 3 summarizes the results, reporting the average oracle values across 10 seeds with standard error and the highest value achieved within any seed (in parentheses). Since our goal is to assess optimization performance across all input scaffold molecules, the average oracle value of the resulting molecules after optimization serves as our primary benchmark metric. The results demonstrate that CLaSMO consistently achieves competitive performance, frequently surpassing or equaling baseline methods in both average and maximum oracle values. For instance, in the albuterol_similarity task, CLaSMO achieved the highest average score (0.334) and maximum score (0.667), surpassing all baselines. Similarly, in the menstranol_mpo task, CLaSMO attained an average score of 0.254, demonstrating notable improvements in multi-property optimization. CLaSMO also excelled in the zaleplon_mpo task, which is one of the most challenging optimization tasks in this benchmark, providing the best average and maximum scores among all methods.

Overall, CLaSMO consistently delivers the best or highly competitive results, and the baseline models often require more evaluations to achieve comparable outcomes. Specifically, CLaSMO achieved the highest average oracle values in 11 out of 20 tasks and ranked in the top 2 across 19 tasks. Moreover, it identified the overall maximum oracle value in 10 out of 20 tasks and placed in the top 2 in 16 tasks. Additionally, we report the Top-10 average oracle values identified by each methodology at each optimization step in Fig. 9, offering another perspective on the performance of different methodologies. This figure demonstrates that CLaSMO not only identifies the highest values faster than other methodologies but also achieves the highest overall Top-10 average in the majority of optimization tasks. Even in cases where baseline methodologies

---

[7]We provide further details on experimental setting including hyperparameter selection on Appendix A.5.

attain a higher Top-10 average value by the end of 100 iterations, such as in celecoxib_rediscovery, gsk3b, and thiothixene_rediscovery, CLaSMO demonstrates superior performance during the initial phases of optimization. These results further reinforce our claims regarding the sample-efficiency of CLaSMO, establishing it as a cost-effective optimization methodology for low-budget scenarios in real-life settings.

Among the competitor methods, Graph-GA emerged as a strong contender, identifying the highest overall oracle values in 7 out of 20 tasks and achieving the overall maximum oracle value in 7 tasks. One notable characteristic of genetic algorithm-based approaches compared to other benchmark methodologies is that they do not inherently preserve the scaffold structure, as they rely on mutation and crossover-based modifications. Consequently, while Graph-GA demonstrates strong performance, its scaffold-optimization outcomes reflect a different methodological focus, making direct comparisons with other approaches less straightforward[8].

Overall, our findings in this extensive benchmark setting highlight CLaSMO's ability to consistently identify molecules with superior properties compared to baseline methodologies, all within the same number of oracle evaluations. This demonstrates its effectiveness across optimization tasks of varying complexity. By leveraging its conditioning and LSBO-based search mechanisms, CLaSMO generates substructures precisely tailored to the atomic environment, significantly enhancing sample efficiency and ensuring robust performance across a diverse range of tasks.

## 5.5 Ablation Study

We conducted an ablation study to evaluate the impact of incorporating atomic environment features as condition vectors in the generative process. In the training of the baseline VAE model, only substructures of the molecules were used, without conditioning on atomic properties. This setup allowed us to isolate the effect of conditioning on scaffold optimization.

We repeated the QED optimization experiments using the VAE model. The QED optimization results, shown in Table 4, demonstrate that incorporating atomic environment features in the CVAE significantly improves the generation of substructures. The conditioning mechanism enhances the model's ability to generate substructures that bond more effectively with the scaffold, leading to higher success rates in optimizing molecular properties. While the VAE without conditioning still gains from the LSBO framework's efficient optimization, incorporating atomic properties as conditions significantly enhances the quality of the generated molecules.

## 5.6 Interactive Optimization

In CLaSMO, the modification region of the input scaffold is typically selected during the automated optimization process. However, the framework also supports an interactive mode, where a chemical expert manually selects the region of the molecule to modify. In this mode, the expert identifies the specific atom or region for modification, rather than relying on CLaSMO's automated selection. Once the region is chosen, the rest of the process remains unchanged—CLaSMO continues to optimize the molecule by generating substructures using the CVAE and refining them through LSBO to improve target properties.

This interactive approach offers several key advantages. It enables the integration of expert knowledge into the optimization process, allowing CLaSMO to operate in a Human-in-the-loop setting. Experts can leverage their domain-specific insights to target specific regions they find promising, ensuring that modifications are not only data-driven but also aligned with scientific understanding. Meanwhile, CLaSMO maintains its efficient optimization process, using LSBO to enhance molecular properties and preserve sample efficiency. Detailed user guidelines for this open-source web application are provided in the Appendix A.5.

---

[8]As outlined in Appendix A.5, we addressed this issue by fine-tuning the mutation rate within these methodologies to reduce significant deviations from input scaffold.

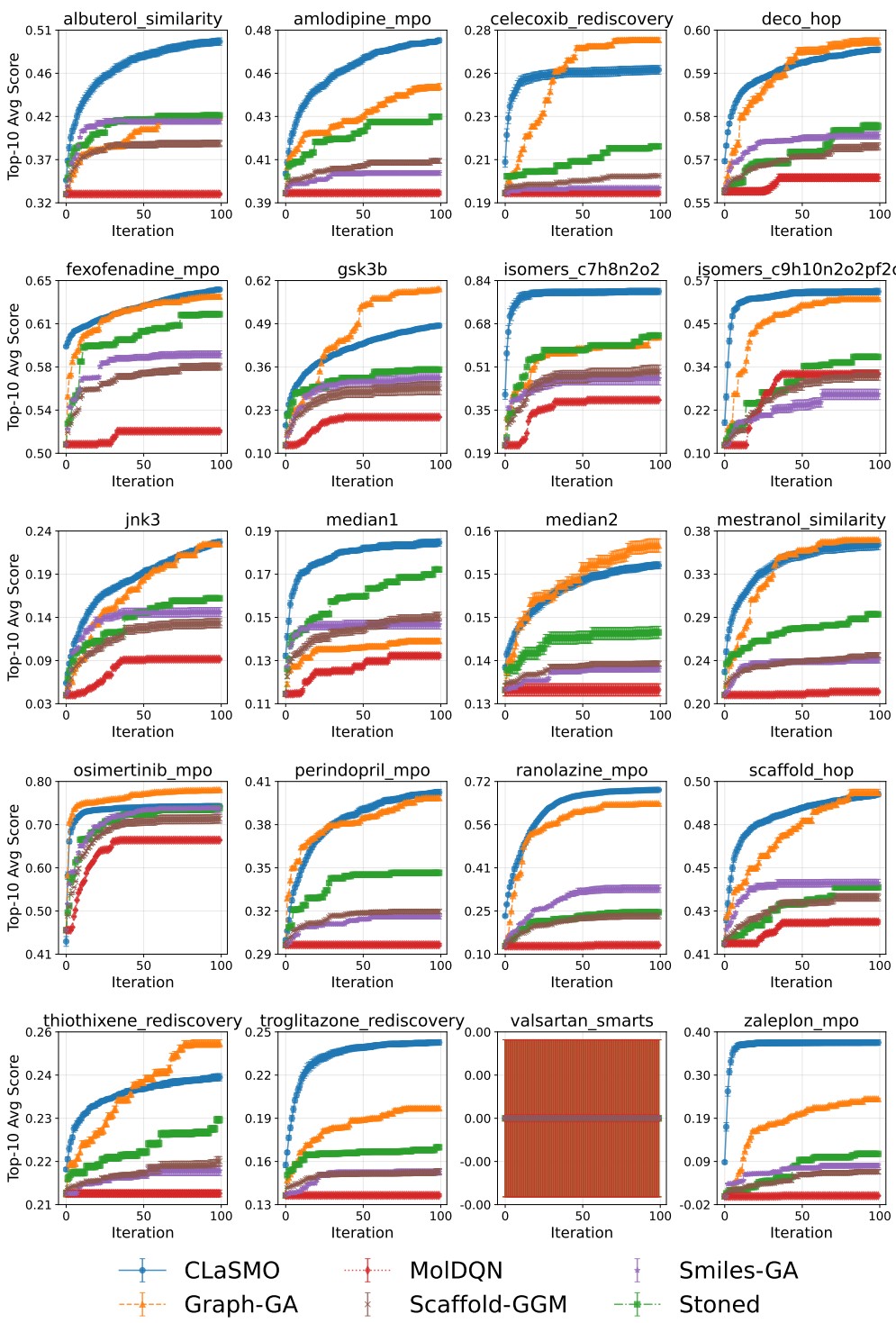

Figure 9: Comparison of Top-10 average values obtained at each iteration for CLaSMO, Smiles-GA, Graph-GA, Stoned, MolDQN, and Scaffold-GGM. CLaSMO identifies molecules with higher Top-10 values faster, particularly in the initial iterations, as seen in tasks like albuterol_similarity, isomers_c7h8n2o2, isomers_c9h10n2o2pf2cl, median1, and zaleplon_mpo, and achieves the highest overall Top-10 average score after 100 iterations in many optimization tasks. Among benchmark methodologies, Graph-GA is highly competitive, followed by Stoned, Smiles-GA, Scaffold-GGM, and MolDQN.

Table 3: Average values of optimization results across 10 seeds with 100 oracle evaluations. The maximum result obtained is reported in parentheses. Input column denotes the average oracle value of the input scaffolds.

| Oracle | Input | CLaSMO | Smiles-GA | Graph-GA |
|---|---|---|---|---|
| albuterol_similarity | 0.186 | **0.334**±0.007 (**0.667**) | 0.248±0.009 (0.451) | 0.301±0.009 (0.424) |
| amlodipine_mpo | 0.138 | 0.234±0.015 (0.464) | 0.144±0.013 (0.440) | **0.302**±0.015 (**0.483**) |
| celecoxib_rediscovery | 0.088 | 0.128±0.007 (**0.328**) | 0.093±0.007 (0.225) | **0.157**±0.008 (0.327) |
| deco_hop | 0.509 | 0.552±0.003 (0.655) | 0.520±0.002 (0.623) | **0.568**±0.0034 (**0.675**) |
| fexofenadine_mpo | 0.090 | **0.532**±0.007 (0.700) | 0.191±0.022 (0.643) | 0.398±0.03 (**0.713**) |
| gsk3b | 0.025 | **0.227**±0.012 (0.650) | 0.107±0.010 (0.870) | 0.181±0.02 (**0.950**) |
| isomers_c7h8n2o2 | 0.026 | 0.209±0.020 (**0.967**) | 0.084±0.014 (0.819) | **0.228**±0.029 (0.883) |
| isomers_c9h10n2o2pf2cl | 0.014 | **0.155**±0.019 (**0.788**) | 0.046±0.008 (0.417) | 0.145±0.016 (0.636) |
| jnk3 | 0.010 | **0.092**±0.005 (0.340) | 0.046±0.004 (0.200) | 0.086±0.007 (0.270) |
| median1 | 0.050 | **0.108**±0.004 (**0.235**) | 0.067±0.004 (0.177) | 0.097±0.002 (0.160) |
| median2 | 0.076 | 0.096±0.004 (0.181) | 0.080±0.003 (0.157) | **0.100**±0.004 (**0.199**) |
| mestranol_similarity | 0.094 | **0.254**±0.005 (**0.466**) | 0.117±0.007 (0.306) | 0.211±0.01 (0.418) |
| osimertinib_mpo | 0.071 | 0.664±0.010 (0.786) | **0.687**±0.013 (0.812) | 0.675±0.033 (**0.817**) |
| perindopril_mpo | 0.086 | 0.142±0.014 (**0.466**) | 0.111±0.011 (0.378)) | **0.210**±0.015 (0.419) |
| ranolazine_mpo | 0.020 | **0.544**±0.010 (**0.733**) | 0.066±0.010 (0.453) | 0.304±0.026 (0.680) |
| scaffold_hop | 0.351 | **0.416**±0.005 (0.492) | 0.364±0.004 (**0.543**) | 0.393±0.002 (0.522) |
| thiothixene_rediscovery | 0.115 | 0.146±0.006 (0.278) | 0.120±0.007 (0.243) | **0.163**±0.007 (**0.281**) |
| troglitazone_rediscovery | 0.087 | **0.164**±0.005 (**0.352**) | 0.095±0.004 (0.190) | 0.127±0.005 (0.238) |
| valsartan_smarts | 0.000 | 0.000±0.000 (0.000) | 0.000±0.000 (0.000) | 0.000±0.000 (0.000) |
| zaleplon_mpo | 0.000 | **0.129**±0.010 (**0.497**) | 0.008±0.004 (0.300) | 0.050±0.008 (0.324) |

| Oracle | Stoned | MolDQN | Scaffold-GGM |
|---|---|---|---|
| albuterol_similarity | 0.281±0.009 (0.495) | 0.248±0.004 (0.364) | 0.250±0.005 (0.410) |
| amlodipine_mpo | 0.200±0.011 (0.428) | 0.234±0.015 (0.464) | 0.155±0.012 (0.422) |
| celecoxib_rediscovery | 0.112±0.006 (0.24) | 0.104±0.005 (0.225) | 0.097±0.006 (0.260) |
| deco_hop | 0.519±0.002 (0.625) | 0.527±0.002 (0.625) | 0.530±0.003 (0.625) |
| fexofenadine_mpo | 0.216±0.024 (0.662) | 0.376±0.007 (0.601) | 0.170±0.015 (0.640) |
| gsk3b | 0.128±0.009 (0.450) | 0.129±0.004 (0.290) | 0.090±0.007 (0.350) |
| isomers_c7h8n2o2 | 0.180±0.020 (0.741) | 0.178±0.011 (0.549) | 0.120±0.015 (0.700) |
| isomers_c9h10n2o2pf2cl | 0.076±0.011 (0.444) | 0.076±0.010 (0.495) | 0.040±0.009 (0.400) |
| jnk3 | 0.061±0.005 (**0.400**) | 0.053±0.002 (0.110) | 0.063±0.003 (0.170) |
| median1 | 0.100±0.004 (0.217) | 0.086±0.002 (0.160) | 0.070±0.003 (0.200) |
| median2 | 0.091±0.004 (0.170) | 0.080±0.003 (0.157) | 0.082±0.004 (0.160) |
| mestranol_similarity | 0.141±0.008 (0.418) | 0.122±0.005 (0.248) | 0.150±0.006 (0.350) |
| osimertinib_mpo | 0.231±0.027 (0.795) | 0.613±0.003 (0.708) | 0.300±0.020 (0.760) |
| perindopril_mpo | 0.109±0.009 (0.378) | 0.131±0.012 (0.390) | 0.110±0.011 (0.400) |
| ranolazine_mpo | 0.051±0.008 (0.340) | 0.028±0.004 (0.211) | 0.040±0.006 (0.320) |
| scaffold_hop | 0.362±0.004 (0.529) | 0.378±0.002 (0.522) | 0.370±0.004 (0.530) |
| thiothixene_rediscovery | 0.135±0.007 (0.245) | 0.122±0.006 (0.243) | 0.120±0.007 (0.260) |
| troglitazone_rediscovery | 0.113±0.004 (0.189) | 0.100±0.002 (0.154) | 0.105±0.003 (0.200) |
| valsartan_smarts | 0.000±0.000 (0.000) | 0.000±0.000 (0.000) | 0.000±0.000 (0.000) |
| zaleplon_mpo | 0.012±0.004 (0.350) | 0.0002±0.0001 (0.006) | 0.050±0.006 (0.400) |

## 6    Conclusion

In this paper, we introduced CLaSMO, a novel framework that combines CVAE and LSBO for scaffold-based molecular optimization. Our approach efficiently explores latent spaces to optimize molecular properties, such

Table 4: QED optimization results: Comparison of VAE and CVAE at different $\tau$ thresholds, reporting mean and max improvement rates.

| Metric | $\tau = 0.25$ | $\tau = 0.50$ |
|---|---|---|
| **Mean Imp. (VAE)** | 12.31% | 8.41% |
| **Mean Imp. (CVAE)** | 19.14% | 16.08% |
| **Max Imp. (VAE)** | 68.16% | 32.28% |
| **Max Imp. (CVAE)** | 114.57% | 74.74% |

as QED and docking scores, while maintaining structural similarity with the input scaffold to improve the chances of real-world viability of optimized molecules. By conditioning substructure generation on the atomic environment of the target region in the input molecule, CLaSMO generates chemically meaningful modifications. The experimental results—encompassing QED optimization, docking simulation score optimization, and 20 additional benchmark property optimization tasks—demonstrate that CLaSMO consistently delivers superior performance across diverse settings and optimization objectives of varying complexity, even with limited training data. By utilizing a significantly smaller model compared to many existing methods and exhibiting superior sample efficiency, CLaSMO proves to be a highly effective methodology for low-budget molecular optimization scenarios. Furthermore, CLaSMO's ability to control structural divergence through similarity constraints ensures robust performance across different optimization tasks. Although this paper focuses on scaffold-based modifications, CLaSMO is fully compatible with whole molecules, requiring no changes to its methodology. Additionally, we have open-sourced a web application to allow chemical experts to use CLaSMO in a Human-in-the-Loop setting, further extending its practical applicability. Overall, CLaSMO exemplifies the power of combining scaffold-based strategies with LSBO, offering a highly effective tool for targeted drug discovery and broader molecular design challenges.

### Broader Impact Statement

The work presented in this paper has the potential to accelerate the discovery of new chemical compounds, which can positively impact various industries, particularly pharmaceuticals and materials science. By improving the efficiency of molecular optimization, CLaSMO could contribute to the development of more effective drugs, especially in regions facing significant public health challenges, such as the need for rapid vaccine development. Moreover, CLaSMO's focus on real-world applicability increases the chances that the compounds discovered are not just theoretical but can be realistically produced, which is critical for translating scientific innovation into real-world solutions. Moreover, CLaSMO's ability to work in a Human-in-the-Loop setting enables domain experts to directly contribute to the optimization process, enhancing collaboration between artificial intelligence and human expertise.

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

# A  Appendix

## A.1  Definitions of Atomic Features

In Table 5, we provide the definitions of the six atomic features we utilized in our data preparation setting.

| Property | Description |
|---|---|
| Atom Type | Specifies the element (e.g., carbon, oxygen), which determines bonding capabilities and chemical behavior. |
| Hybridization | Describes the mixing of atomic orbitals, influencing shape, bond angles, and bonding interactions. |
| Valence | Refers to the number of bonds an atom can form, indicating potential for additional bonding. |
| Formal Charge | Represents the charge if all bonding electrons are shared equally, crucial for reactivity and bonding sites. |
| Degree | Denotes the number of directly attached atoms (neighbors), providing insight into the local atomic environment. |
| Ring Membership | Indicates if the atom is part of a ring structure, impacting substructure rigidity, stability, and bonding behavior. |

Table 5: Key atomic properties used in CVAE training in CLaSMO framework.

## A.2  QED Value Distribution of Input Substructures

Figure 10 shows the distribution of QED values for the substructures used to train the CVAE model, generated through our data preparation process.

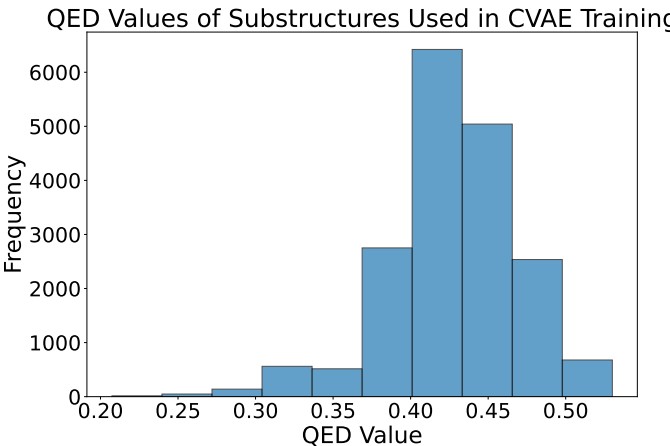

Figure 10: QED value distribution of substructures derived from our data preparation methodology.

### A.3 Condition Vector Embeddings

The simplicity of our dataset allowed us to reduce the latent space to two dimensions without compromising reconstruction quality, thereby enhancing LSBO performance. Consequently, the decision to use 2-dimensional embeddings for the condition vectors, rather than the original six-dimensional features, was motivated by this low-dimensional latent space configuration in our CVAE model.

Using a six-dimensional condition vector in a CVAE with a 2-dimensional latent space poses a significant challenge: the decoder may overly rely on the condition vectors during reconstruction, potentially diminishing the latent space's ability to encode critical structural information. This can result in a loss of critical structural information within the latent space, as a disproportionate amount of information is drawn from the condition vectors. Such an imbalance can degrade the quality of the latent space, negatively affecting the surrogate model used in LSBO and ultimately hindering the optimization process. Conversely, when 2-dimensional condition vectors are used, the latent vectors must encode more information about the input instances, as the decoder's reliance on the condition vectors is reduced. This shift encourages the latent space to better capture the underlying structural properties, improving its utility for LSBO. In LSBO, latent vectors are utilized to train the surrogate model, which guides the search process by predicting target property values within specific regions of the latent space. Therefore, ensuring that the latent vectors are information-rich is crucial for achieving efficient LSBO performance.

To evaluate this effect, we compared CVAE models trained with 2-dimensional embeddings of the conditional features to those trained with the original 6-dimensional features. For both models, we obtained the 2-dimensional latent vectors of input scaffolds using the encoders of the trained CVAEs and calculated the corresponding QED values. We then trained random forest regressors to predict the QED values of the input scaffolds based on the latent vectors from each model. Our results showed that the CVAE with 2-dimensional condition embeddings achieved a 5% lower mean squared error in QED predictions, underscoring the benefits of reduced-dimensional embeddings in preserving latent space quality and improving LSBO performance.

We acknowledge that the model with 6-dimensional condition vectors converged faster and achieved slightly better reconstruction accuracy during its training compared to model with 2-dimensional condition vector embeddings. However, our primary goal was to design the latent space for LSBO, where the latent representation provided by the 2-dimensional embeddings proved more effective.

### A.4 Model and Hyperparameter Selection

In chemical VAE models, it has been established that setting the KL divergence weight $\beta < 1$ can improve generative performance Yan et al. (2020), and our findings are consistent with this. We experimented with a range of $\beta$ values from 1 to $1^{-7}$, selecting models based on their reconstruction accuracy on the training set.

Interestingly, we found that even at very low $\beta$ values, the CVAE retained its generative capacity. However, as $\beta$ increased, we observed a decline in both reconstruction accuracy and the diversity of generated molecules.

To ensure robust training, we applied early stopping in combination with a learning rate scheduler (using PyTorch's ReduceLROnPlateau function). Models were evaluated based on their reconstruction performance and generative diversity, leading us to select the model with $\beta = 0.000001$ as the optimal candidate. Additionally, our CVAE leverages conditional batch normalization De Vries et al. (2017), which improves the impact of conditioning in the generative process.

For CLaSMO's penalization terms, we opted for a straightforward approach rather than an exhaustive hyperparameter optimization process. We set $\lambda_1 = -5$ to penalize cases where the similarity constraint $\text{DICE}(\boldsymbol{S}, \boldsymbol{S'}) > \tau$ was violated. Similarly, we assigned $\lambda_2 = -7.5$ for situations where the generated substructure could not be added to the input scaffold. These values were chosen to assign poor scores in cases where the sampled region did not meet the desired criteria, allowing LSBO to learn that the region is suboptimal. This approach helps guide the optimization process away from unproductive regions and toward more promising areas.

### A.5 Sample-Efficiency Benchmark Experimental Setting

In the sample-efficiency benchmark experiments adopted from Gao et al. (2022), we reduced the number of optimization iterations from 10,000 to 100 to create a low-budget experimental setting. In the original open-source implementation, Smiles-GA uses a population size of 50, while Graph-GA uses a population size of 120. Given our limited optimization budget of only 100 iterations, these values required adjustment. Furthermore, Smiles-GA allows for 500 mutations in its default configuration, and Graph-GA employs a mutation rate of 6.7%. These values are disproportionately high for our experimental setting and context, as they lead to significant divergence from the input scaffold, effectively turning the task into a from-scratch-generation problem. This shift results in an unfair comparison by deviating from the intended optimization context and giving genetic algorithm-based methods an advantage, as other methodologies like CLaSMO, Scaffold-GGM, and MolDQN are constrained to build upon the given scaffold.

To address these challenges, we adjusted the mutation parameters. Specifically, we set the mutation parameter to 10 in Smiles-GA and reduced the mutation rate in Graph-GA to 0.1% to discourage significant deviations from the input scaffold. For the population size, we experimented with values of 5 and 10 in both Smiles-GA and Graph-GA. A population size of 5 yielded superior results for both; therefore, we reported our findings based on this configuration in Table 3.

The Stoned methodology, although categorized as a genetic algorithm in Gao et al. (2022), relies on random rearrangements of atoms within the molecule and lacks a specific hyperparameter to control the degree of divergence from the initial scaffold. Consequently, similar to MolDQN and Scaffold-GGM, we used the default hyperparameters for the Stoned methodology in our experiments. For the CLaSMO experiments, we used a similar setting as in the QED experiments in Section 5.2 .

### Human-in-the-Loop via Web-Application

In Fig. 11, we showcase a sequence of screenshots from our web application, demonstrating the process of molecule optimization. First, the user inputs a SMILES Weininger (1988) string of the chemical compound into the designated text field. Once the input is provided, the application automatically computes the molecule's QED value and generates a visual representation. Subsequently, the user selects a region of interest by drawing a rectangle around the area they wish to modify. Upon confirming the selection, the CLaSMO optimization process is initiated, targeting improvements in the selected molecular region. Upon completion, the optimized molecule is displayed, and the process can be continued by using the resulting molecule as input for further iterations. By incorporating user input in the region selection, we create a Human-in-the-Loop optimization workflow.

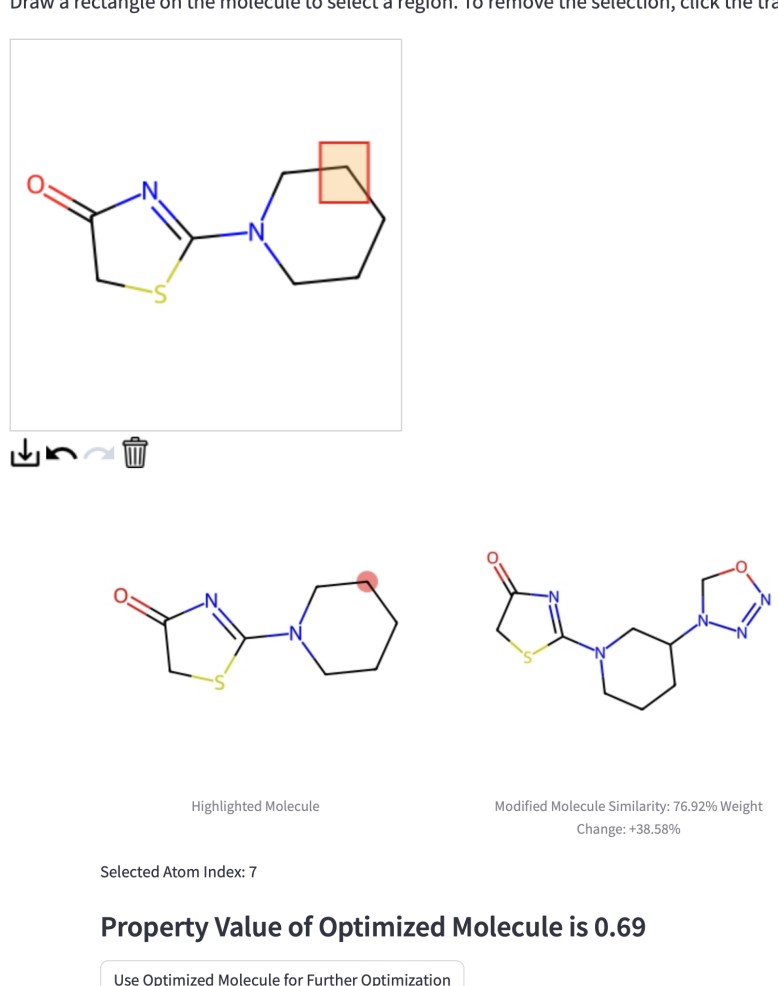

Figure 11: Screenshots from our web application showing the step-by-step process of molecule optimization using CLaSMO in an interactive setting. The process includes inputting a SMILES string, visualizing the molecule's QED value, selecting a region for modification, and initiating the optimization procedure. In this example, the QED value of the input molecule is improved from 0.56 to 0.69, where the resulting molecule is demonstrated in the bottom-right figure.

