# OpenReview forum: "Conditional Latent Space Molecular Scaffold Optimization for Accelerated Molecular Design"
_TMLR — Rejected by TMLR_

### Review · Reviewer_P2E2 · 2024-12-09

**Summary Of Contributions:**

The authors propose Conditional Latent Space Molecular Scaffold Optimization (CLaSMO) for optimizing molecules for improved property with a given seed molecule. The work changes the VAE used in Gómez-Bombarelli et al. (2018) to CVAE, to enable adding new substructure conditioned on an initial scaffold structure.

**Audience:**

Yes

**Claims And Evidence:**

Yes

**Requested Changes:**

- Please compare with other molecular optimization baselines such as Genetic Algorithm [2], and selected methods used in Practial Molecular Optimization benchmark (PMO) [1] - which can also be amenable to molecular optimization with a given seed.
- Evaluate on more numbers of challenging oracles such as ones in PMO [1].

[1] Gao, Wenhao, Fu, Tianfan, Sun, Jimeng, and Coley, Connor W. (2022). Sample Efficiency Matters: A Benchmark for Practical Molecular Optimization.

[2] Tripp, Austin. (2023) Genetic algorithms are strong baselines for molecule generation. https://portal.valencelabs.com/blogs/post/genetic-algorithms-are-strong-baselines-for-molecule-generation-yy7JdogPQ5nadJO

**Strengths And Weaknesses:**

Strengths:

- Sample efficient molecular optimization is very useful for real-world drug design settings. The approach of doing Bayesian Optimization of latent space is technically sound, and well-motivated for improving sample efficiency.
- Ablation studies show that incorporating atomic environment features using the proposed CVAE architecture improves the generation of substructures versus unconditional VAE.
- An open source web-application is provided for users to easily examine the output of the model.

Weaknesses:
- The evaluation in this paper is insufficient, as it benchmarks against baselines on a single, and trivial oracle (QED). As highlighted in [1], "QED is likely to have a global maximum of 0.948, and even random sampling could reach that value. It is disabled to meaningfully distinguish different algorithms." This makes QED an inadequate choice for evaluating the relative performance of molecular optimization algorithms.
- While Section 5.3 provides evaluation of the method on docking oracles, the performance is not compared against any other baseline, so it’s difficult to evaluate the effectiveness of the proposed model.
- In Table 2, comparison done is with other methods on the amount of training data, however such comparison is misleading because methods such as JT-VAE only requires unlabelled chemical structures and these are easily obtainable. It’s more valid to compare against molecular optimization methods on the number of oracle calls, because these tend to be the limiting factor.
- This method only conditions on the features of the single atom (which will connect to a new substructure), but not other parts of the scaffold structure.  This assumption is not true for more challenging tasks such as SBDD, because the substructure added should depend on the entire seed molecule.

Minor points:
- (pg 5) Directly optimizing over all possible … -> Directly iterating over all possible …
- Chemical diversity, a key metric to ensure the model could capture multiple modes which satisfy the similarity constraint and property scores, is not reported in the paper.

---

> ### Author Response · Authors · 2025-01-09
> **Added Experiments with PMO Benchmark and Reflection on Comments**
>
> Thank you for your constructive feedback and for recommending valuable papers in the field. We acknowledge your concerns regarding the lack of empirical results and insufficient benchmarking, and we have taken steps to address these points. We believe the additional experiments effectively address these key issues. In the updated manuscript, you will see the revised parts in red.
>
> ## Weakness 1
> > The evaluation in this paper is insufficient, as it benchmarks against baselines on a single, and trivial oracle (QED). As highlighted in [1], "QED is likely to have a global maximum of 0.948, and even random sampling could reach that value. It is disabled to meaningfully distinguish different algorithms." This makes QED an inadequate choice for evaluating the relative performance of molecular optimization algorithms.
> ### Our Response
> We appreciate the concern about relying solely on QED and agree it’s insufficient. Although we initially chose QED due to its wide use, we recognize its trivial nature. To address this, we evaluated CLaSMO on 20 PMO benchmark tasks with 100 oracle evaluations each, where it outperformed benchmarks, demonstrating superior sample efficiency as competitors required more iterations to match its performance.
>
> ## Weakness 2
> > While Section 5.3 provides evaluation of the method on docking oracles, the performance is not compared against any other baseline, so it’s difficult to evaluate the effectiveness of the proposed model.
> ### Our Response
> We understand the reviewer’s concern regarding the absence of baseline comparisons for docking score optimization. The primary reason for this limitation was the significant computational cost associated with our high-accuracy docking simulations, which were performed using Schrödinger Software. Unlike RDKit-like calculations, these simulations are computationally intensive, requiring minutes to hours per evaluation. Consequently, we focused on demonstrating that CLaSMO can improve docking scores effectively, and we refer to new experiments in Section 5.4 for comprising against other baselines.
>
> ## Weakness 3
> >  In Table 2, comparison done is with other methods on the amount of training data, however such comparison is misleading because methods such as JT-VAE only requires unlabelled chemical structures and these are easily obtainable. It’s more valid to compare against molecular optimization methods on the number of oracle calls, because these tend to be the limiting factor.
> ### Our Response
> - We appreciate the comment and agree that relying solely on training data size can be misleading. As with JT-VAE, our CVAE doesn’t require labeled property data (e.g., QED) during training, which is only used in optimization.
> - We included training data sizes for completeness, but because oracle evaluations weren’t consistently reported in some studies, we focused on dataset sizes. With new experiments, we now compare performance using a fixed number of oracle evaluations, offering multiple perspectives.
>
> ## Weakness 4
> > This method only conditions on the features of the single atom (which will connect to a new substructure), but not other parts of the scaffold structure. This assumption is not true for more challenging tasks such as SBDD, because the substructure added should depend on the entire seed molecule.
> ### Our Response
> Thank you for your fair criticism. Although our method focuses on single-atom features, these inherently reflect local context. For example, degree represents the number of neighboring atoms directly bonded to the target atom, valence captures how those neighbors shape the atom’s bonding capacity, and ring membership indicates whether the atom lies within a locally constrained ring structure. By highlighting these localized yet context-aware features, our approach effectively guides substructure generation. We acknowledge that more complex tasks, like SBDD, would benefit from global structural features, which are currently absent but could be integrated in future work. Nonetheless, our experiments show that the present features suffice for the optimization tasks in this study.
>
> ## Actions for Requested Changes
> Thank you very much for suggesting the PMO benchmark in [1] and informing us about Genetic Algorithm discussion provided in [2]. As we discussed in our response to the Weakness 1, we included new sets of experiments to evaluate the sample-efficiency of CLaSMO against Genetic Algorithm based (Smiles-GA and Stoned), Reinforcement Learning Based (MolDQN), and Graph-Generative-Model (Scaffold-GGM) based approaches in Section 5.4.
>
> ## Additional Remarks
> Thank you for highlighting chemical diversity. While it’s an important measure, we focused on minor scaffold modifications rather than generating compounds from scratch, therefore generated molecule remains close to the seed molecule, and for this reason, we decided not to focus on diversity.

---

> ### Comment · Reviewer_P2E2 · 2025-01-09
>
> Thank you for addressing many of my concerns. I appreciate the author for adding the evaluation on 20 PMO benchmark tasks. It’s encouraging to see the promising result of CLaSMO. However, I still have 2 remaining concerns over the updated  evaluation.
>
> One thing I noticed is that the budget for oracle calls has decreased from 10000 in PMO [1]  to just 100 in CLaSMO. I certainly believe there's a use case for molecular optimization models operating in a very small number of oracle calls (~100), especially in the applications involving wet applications. However, it would still be nice to have a discussion on the sample efficiency of CLaSMO over different oracle budgets ranges (0 - 10000) compared to other methods. Furthermore, it’s preferable to have a plot of optimization curve, with y-axis being the top-10 average of the property score and x axis being the number of the oracle calls (from 0 to 10000).
>
> Secondly, I am concerned if the hyperparameters of the baseline genetic algorithm are not changed according to the decrease in the oracle budget. For example, by default the population size for genetic algorithms may be set to 100 - and if the budget for oracle calls is only 100, no learning will occur. The population size should be changed accordingly (for example 5) if we only have a budget of 100 oracle calls. It would be ideal to see these details in the appendix.
>
> Furthermore, since CLaSMO operates on the graph level. The Graph GA - one of the top methods from PMO should be included in the ablation.

---

> ### Author Response · Authors · 2025-01-14
> **Oracle Calls, Focus of CLaSMO, and Hyperparameters**
>
> Thank you for your kind feedback and for highlighting these important points. We appreciate the opportunity to clarify and improve our work further. Below, we address the remaining concerns in detail.
>
> ## Oracle Budget of 100 and Clarification of Methodology Focus
>
> In our study, we intentionally set the oracle budget to 100 iterations, significantly lower than the 10,000 iterations used in PMO [1]. This decision reflects the primary focus of our methodology, which is specifically designed for low-budget optimization problems requiring high sample efficiency. These scenarios are particularly relevant for applications involving costly and time-consuming wet-lab experiments, where minimizing oracle calls is crucial.
>
> We acknowledge that our previous manuscript versions may not have sufficiently emphasized this focus. In the revised manuscript, we have explicitly clarified that our methodology is tailored for resource-constrained settings, and we expanded the discussion on this aspect in the introduction and conclusion sections. Additionally, we prepared the Top-10 average plots based on 100 iterations, as shown in Figure 9, to illustrate our methodology's performance under this constrained budget.
>
> While we acknowledge that exploring the sample efficiency of CLaSMO across a broader range of oracle budgets (e.g., 0–10,000 iterations) would provide valuable insights, this falls beyond the current scope of our study, which is specifically focused on low-budget scenarios. CLaSMO was designed to address practical needs in resource-constrained settings, such as wet-lab experiments, where minimizing the number of oracle calls is critical.
>
> We do not claim that CLaSMO would continue to deliver better performance compared to baseline models as the number of iterations is increased to 10,000. Baseline methods may achieve superior results in such high-budget scenarios. Instead, our study emphasizes CLaSMO's capability to achieve competitive performance efficiently within a limited number of oracle calls, which is often the key consideration in real-world applications. The trends observed within 100 iterations demonstrate CLaSMO's strength in achieving high sample efficiency, making it particularly suitable for scenarios where computational or experimental costs must be tightly controlled.
>
> ## Graph-GA and Population Size in Genetic Algorithms
>
>
> Thank you for suggesting the inclusion of Graph-GA in our analysis. We have incorporated new results based on Graph-GA into our manuscript.
>
> In our initial results, the population size was set to 50, which we later recognized was not well-aligned with a constrained oracle budget of 100. Following your suggestion, we conducted additional experiments with reduced population sizes of 5 and 10 for both Smiles-GA and Graph-GA. These experiments revealed that a population size of 5 yielded better performance, and we have updated the reported results accordingly in the revised manuscript.
>
> Further details on these hyperparameter adjustments are now included in Appendix A.5, where we describe the experimental setup and rationale in more depth. Additionally, we have revised our results for Smiles-GA, added results from Graph-GA, and corrected calculation errors in the previously reported CLaSMO results.

---

> > ### Comment · Reviewer_P2E2 · 2025-01-14
> >
> > I thank the author for the new experiments and the openness about the details for the baseline comparisons. I agree with the author's point that genetic algorithms do not preserve the scaffold completely - therefore is not directly comparable with proposed CLaSMO. While it may be possible to restrict genetic algorithm to crossovers and mutations which perserve the scaffold structure - such evaluation would definitely be outside of the scope for this work and can be left for future work.
> >
> > CLaSMO's main use case advantage for oracle budgets <100 with the strict requirement of persevering scaffold structure - is certainly a common and reasonable use case for the real world setting. This point is clear now and I'll update my evaluation.

---

### Review · Reviewer_qYeF · 2024-12-11

**Summary Of Contributions:**

The authors tackle the important task of generative modeling for molecular design, highlighting that the two main approaches (either modification or from-scratch generation) face challenges relating to sample efficiency for model training and/or real-world feasibility. They train a conditional VAE (including substructures and conditioning on 6-d atomic features) and then afterwards perform a search in the latent space to find viable modifications to the molecular scaffolds which optimise a desired molecular property (QED - Quantitative Estimate of Drug-likeness).  Their method 'CLaSMO' is shown to provide high QED in comparison with alternatives. In addition to this principal contributions, the authors also create an interactive website for drug modification using CLaSMO, as well as a benchmark dataset BRICS which is uniquely designed to facilitate evaluation of their method.

**Audience:**

Yes

**Broader Impact Concerns:**

No concerns.

**Claims And Evidence:**

Yes

**Requested Changes:**

C1. Add a couple of sentences to 4.1 reminding the readers why alternatives to BRICS cannot be used for benchmarks in this application (i.e. not only the features of BRICS, which are more or less there, but highlighting which of these features do not exist elsewhere and therefore specifically what value BRICS adds to the community.

C2. Relatedly - is the BRICS dataset going to be released? It sounds like it would be helpful for the community.

C3 (minor). Last line of page 9 - the information about the data type for the string-based representation might also be well-placed in section 4.1. As a reader I was interested in this information in 4.1 but didn't arrive at it until 5.1.

C4. As I'm sure the authors know, the choice in prior for the latent space dictates the nature of the distributions learned by the VAE. In this example, the authors have chosen the standard isotropic Gaussian. Furthermore, the authors explain that the latent space was 2-dimensional (for both the conditioning information embedding as well as the structural embedding). Given these two (strong) constraints/assumptions, I wonder (a) what the authors believe is being really encoded in these 2 dimensions, and/or whether they are overly compressing information, notwithstanding the 99% reconstruction accuracy, and (b) why the authors believe these latent factors are Gaussian distributed. For the latter point, my experience is that even though a Gaussian can be sufficient for reconstruction, it can lead to issues in latent space traversal or generation. There are a huge range of VAE variations exploring the potential of different priors, and I'd appreciate a comment on the impact of these decisions.

C5 (minor). There are a few acronyms that it wouldn't hurt to re-define on a section-basis. e.g. the section 5.2 title, QED can be written in full. For readers outside the domain (like myself) it's a little inefficient to have to keep checking for definitions.

C6 (minor). In section 4.3 it says 'we perform a targeted search' - it would be helpful to remind the reader that this is performed <after> the CVAE has been trained (as opposed to during).

C7. The citation formatting was a bit lacking (sometimes in parentheses, sometimes lacking parentheses) - easy to fix.

C8. I'd appreciate a comment about the efficiency parts.

**Strengths And Weaknesses:**

S1. Firstly, although I have spent a lot of time on VAEs and generative models, I have not done so in the domain of molecular optimisation, and so this application is relatively unknown to me. That said, I felt that the authors did a fantastic job of making this topic accessible to unfamiliar readers. Even though I cannot comment on how the authors' work compares to alternatives in this particular area, or what the standards are with respect to evaluations, at no point did I feel uncomfortable with the topic area, and in my view this is testament to the clarity of explanation.

S2. Secondly, I appreciate the multi-dimensional nature of the contribution -  assuming reviewers more familiar with the application domain are satisfied with the evaluations and comparisons the authors undertook (and I defer to their judgement in this area), I applaud the authors for not only submitting a new method, but also working on a bespoke benchmark for evaluation, as well as the website for the interactive modification of molecules. This makes the contribution potentially valuable for multiple reasons.

S3. Whilst I am not familiar with the standards for evaluation in this domain, it felt like the authors did a good job of evaluating their method against alternatives and across a range of metrics/perspectives.

W1. Overall, there are some clarifications which I would appreciate (see requested changes), and I defer to other reviewers' expertise regarding the standards / expectations in the domain of molecule generation.

W2. The authors refer to efficiency as one of the principal advantages of their method. However, I didn't really feel like this element was strongly tested or evaluated. I think either the authors can make a causal argument for this advantage in the discussion, or it needs to be addressed more formally (e.g. testing the sensitivity of the results to different dataset sizes).

---

> ### Author Response · Authors · 2025-01-09
> **Added Experiments, CLaSMO Design Choices and Reflection on Comments**
>
> Thank you for your constructive feedback. We acknowledge the concerns raised, particularly regarding our prior selection of the CVAE and the criticism about sample efficiency, and we have taken steps to address them thoroughly. We believe the additional experiments resolve the key issues, and our responses provide a justification for our design choices. In the updated manuscript, you will see the revised parts in red.
>
> ## Weakness 1
> > Overall, there are some clarifications which I would appreciate (see requested changes), and I defer to other reviewers' expertise regarding the standards / expectations in the domain of molecule generation.
> ### Our Response
> Thank you for providing detailed set of questions and points where clarifications needed. We answered them one-by-one below on the Actions for Requested Changes section.
>
> ## Weakness 2
> > The authors refer to efficiency as one of the principal advantages of their method. However, I didn't really feel like this element was strongly tested or evaluated. I think either the authors can make a causal argument for this advantage in the discussion, or it needs to be addressed more formally.
> ### Our Response
> Thank you for this fair criticism. We added experiments on 20 additional optimization tasks, comparing CLaSMO against popular methodologies, and demonstrating its superior sample efficiency. Please see Section 5.4 in the updated manuscript.
>
> ## Actions for Requested Changes
> ### Change 1
> > Add a couple of sentences to 4.1 reminding the readers why alternatives to BRICS cannot be used for benchmarks in this application
>
> Thank you pointing this out. We updated the first paragraph of Section 4.1 to further elaborate on advantage of using BRICS.
>
> ### Change 2
> >  Relatedly - is the BRICS dataset going to be released? It sounds like it would be helpful for the community.
>
> Yes, the dataset and the Python code for dataset preparation are included in the revised supplementary materials submitted with this paper. Additionally, we have prepared an open-source GitHub repository that will include these materials along with the code for our web application. This repository will be made publicly available soon.
>
> ### Change 3
> >  Last line of page 9 - the information about the data type for the string-based representation might also be well-placed in section 4.1. As a reader I was interested in this information in 4.1 but didn't arrive at it until 5.1.
>
> Thank you very much for providing this feedback. After reviewing our manuscript, we agreed with the reviewer and updated the last paragraph of Section 4.1.
>
> ### Change 4
> > Prior distribution selection for CVAE.
>
> We chose the isotropic Gaussian prior for its simplicity, widespread use in VAEs, and compatibility with LSBO. This prior ensures a smooth, continuous latent space essential for effective sampling and optimization. Acquisition functions in BO, such as Upper Confidence Bound, assume Gaussianity in their surrogate models, and deviations from this can cause mismatches, hindering optimization. While richer priors like Gaussian mixtures could capture complex distributions, they may create discontinuities or separated clusters in the latent space, making surrogate models less effective. This Gaussian prior worked well in our study, but we acknowledge its limitations for more complex tasks. Future work will explore alternatives to prior selection to further enhance LSBO as we believe it can be an important contribution to the LSBO community.
>
> ### Change 5
> > There are a few acronyms that it wouldn't hurt to re-define on a section-basis. e.g. the section 5.2 title, QED can be written in full.
>
> We understand that such acronyms can decrease the reading experience. We re-defined the QED in Section 5 and also in the title of Section 5.2.
>
> ### Change 6
> > In section 4.3 it says 'we perform a targeted search' - it would be helpful to remind the reader that this is performed <after> the CVAE has been trained (as opposed to during).
>
> We thank the reviewer for pointing this out. We added a short statement to the beginning of Section 4.3 to clarify the fact that the search is conducted after the CVAE model is trained.
>
> ### Change 7
> > The citation formatting was a bit lacking (sometimes in parentheses, sometimes lacking parentheses)
>
> We tried the follow the guidelines of TMLR, which states that "When the authors or the publication are included in the sentence, the citation should not be in parenthesis, using \citet{}..... Otherwise, the citation should be in parenthesis".
>
> ### Change 8
> > I'd appreciate a comment about the efficiency parts.
>
> We introduced new experiments on 20 properties (recommended by Reviewer P2E2) to address sample efficiency (Section 5.4). We fixed black-box evaluations at 100 and compared CLaSMO against several benchmarks. Across these 20 tasks, CLaSMO demonstrated the best overall performance. These results show competitor methods need more iterations to match CLaSMO, highlighting its superior sample efficiency.

---

> ### Comment · Reviewer_qYeF · 2025-01-10
> **Response**
>
> Thanks a lot to the authors for receiving the feedback in the same spirit in which it was intended (i.e. constructively!). I appreciate the responses and the changes.
>
> For the citation point, which I admit is only minor, I should've been clearer. In the introduction, for example, there is this text:
>
> ...on modifying/editing existing compounds using generative models Bradshaw et al. (2019); Lim et al. (2020), reinforcement learning Gottipati et al. (2020), genetic algorithms Jensen (2019), or from the domain experts themselves by trial
> and error (Bemis & Murcko, 1996; Schreiber, 2000; Welsch et al., 2010).
>
> In this snippet there are 4 citation instances and only one of them is 'correct' (i.e. the last one). All the others should be fully in parentheses. It's worth double checking these. I believe what 'using citations in the sentence' means is when the author names themselves are integrated grammatically into the phrase. e.g.  "It was developed by Jones et al. (1999)..."  is correct but  "They developed a technique Jones et al. (1999)..." is incorrect and should be "They developed a technique (Jones et al., 1999)."
>
> For the efficiency point, I appreciate that Table 2 now includes the dataset sizes, although this is not give me a huge amount of information for how the performance really relates to the size. e.g. what are the Top QED across methods when using the same dataset sizes? At the moment I'm not able to really grasp the relationship between sample size and performance from this, other than to note that CLaSMO does well on a dataset which is much smaller than the other methods on much bigger datasets. In other words, how much does the Top QED drop when the other methods are tested on a smaller dataset?
>
> Similarly, when looking through Table 3, I'm not sure how to infer the relationship between performance as a function of dataset size, unless I'm reading it wrong? What I was imagining was a clear demonstration for performance as a function of sample size, and/or demonstration of performance as a function of model size (where the latter is a computational complexity point, rather than data efficiency).  Perhaps this is not normal for the domain, but when I read the updated version I still don't see the efficiency point really clearly demonstrated. I don't mind this per se (I think the method is still a good contribution regardless), but if the authors have something to hand I think it would strengthen the paper. If I have misunderstood/missed something in the way these results are presented, of course I'm happy to be corrected!
>
> Otherwise, it was very helpful for me to see the other reviewers' comments and the associated responses.

---

> > ### Author Response · Authors · 2025-01-14
> > **Citation and Sample-Efficiency**
> >
> > Thank you for your kind comments and further questions. We value the opportunity to further refine our work based on your insights.
> >
> > ## Citations
> >
> > Regarding the citation point, we acknowledge that we may have misinterpreted the author guidelines on citation formatting. After reflecting on your comments, we have reviewed and corrected the relevant instances in our manuscript. If any issues remain, we will address them promptly in subsequent revisions, if applicable. Thank you for bringing this to our attention.
> >
> > ## Table 2
> >
> > > For the efficiency point, I appreciate that Table 2 now includes the dataset sizes, although this is not give me a huge amount of information for how the performance really relates to the size. e.g. what are the Top QED across methods when using the same dataset sizes? At the moment I'm not able to really grasp the relationship between sample size and performance from this, other than to note that CLaSMO does well on a dataset which is much smaller than the other methods on much bigger datasets. In other words, how much does the Top QED drop when the other methods are tested on a smaller dataset?
> >
> > ### Our Response
> >
> > Thank you for your comment. Our study primarily focuses on demonstrating CLaSMO’s strong performance under limited data conditions, rather than directly comparing methods on datasets of the same size. While we included training data sizes for completeness, our aim was to provide context rather than to establish a direct relationship between dataset size and performance. Ideally, the best way to compare methodologies would have been to provide the number of optimization iterations (oracle evaluations) required to achieve the reported values for each method, as this directly reflects sample efficiency. However, these details were not consistently reported in prior studies, limiting our ability to perform such comparisons. In the absence of this information, we used dataset sizes as a proxy to provide perspective, tried to provide as much context as possible given the available sources of information
> >
> > ## Table 3
> >
> > > Similarly, when looking through Table 3, I'm not sure how to infer the relationship between performance as a function of dataset size, unless I'm reading it wrong? What I was imagining was a clear demonstration for performance as a function of sample size, and/or demonstration of performance as a function of model size (where the latter is a computational complexity point, rather than data efficiency). Perhaps this is not normal for the domain, but when I read the updated version I still don't see the efficiency point really clearly demonstrated. I don't mind this per se (I think the method is still a good contribution regardless), but if the authors have something to hand I think it would strengthen the paper. If I have misunderstood/missed something in the way these results are presented, of course I'm happy to be corrected!
> >
> > ### Our Response
> >
> > Thank you for your comment. We appreciate your feedback and the opportunity to clarify our focus and methodology. Our primary goal is to evaluate sample efficiency, which we define as the ability to identify high-quality candidates with a limited number of optimization iterations (oracle evaluations). This focus is different from the reviewer’s interpretation of efficiency as a function of dataset or model size.
> >
> > In real-world applications, especially those involving wet-lab experiments, each oracle evaluation—such as calculating a property, running a docking simulation, or conducting an experimental assay—can be extremely time-consuming and costly. For example, a single wet-lab experiment to validate a candidate molecule's properties can take several days and incur significant expenses. Thus, methodologies like CLaSMO, which can deliver high-quality candidates in fewer oracle evaluations, are critical for resource-constrained scenarios.
> >
> > Our analysis prioritizes demonstrating CLaSMO’s ability to efficiently optimize target properties within a limited budget of oracle evaluations, which is distinct from analyzing performance relative to dataset size or model complexity. While Table 3 presents results under specific conditions, it does not aim to explore performance as a function of dataset or model size, as these aspects are not the central focus of our study.
> >
> > We understand that our emphasize on dataset sizes on Table 2 have caused this confusion. We are sorry for not making this clear in our initial response.  We agree that an analysis of performance relative to dataset and model size would provide additional insights, but it lies beyond the scope of this work. Instead, our contribution lies in addressing the practical need for low-budget optimization methodologies that prioritize minimizing the cost and time associated with oracle evaluations.
> >
> > Thank you for recognizing the value of our contribution, and we hope this clarification helps illustrate why our focus on sample efficiency is particularly relevant in this domain.

---

> > > ### Comment · Reviewer_qYeF · 2025-01-15
> > > **Response**
> > >
> > > Thanks once again to the authors for their positive reception of the feedback!
> > >
> > > For the efficiency point - I thought this might be the case, and as mentioned, I would be happy regardless. That said, i think given that the efficiency evaluation is beyond the scope of the work, the claims about the efficiency of the method should be toned down a bit. Indeed, one of the reasons I was 'chasing' this sample-size vs. performance relationship is precisely because the efficiency point is made strongly in the paper (including the abstract). If such an evaluation is beyond the scope of the paper, and rather you note as more of a 'bonus' that the performance of CLaSMO is good even in the smaller data regimes, then this should be reflected in how the contribution is spoken about.
> > >
> > > Thanks again!

---

### Review · Reviewer_dBZ6 · 2025-01-01

**Summary Of Contributions:**

The paper presents Conditional Latent Space Molecular Scaffold Optimization (CLaSMO), which combines a Conditional Variational Autoencoder (CVAE) with Latent Space Bayesian Optimization (LSBO).

### CLaSMO consists of the following steps:

1- The molecule is broken down into substructures using BRICS algorithm. The BRICS algorithm provides information on each substructure in terms of what kind of bonds they can form, such as the chemical context around the atom that dictates how substructures can bond with other parts of a molecule. The atomic features considered are atom type, hybridization, valence, formal charge,
degree, and ring membership.

2- An autoencoder is trained to encoder the 6 features from step 1 into condition vector embeddings (in the experiments dimension of size 2 was used)

3- A conditional VAE  (CVAE) was trained on top of the substructures obtained from step 1 and the conditional embeddings obtained from step 2. During generation, the decoder of the CVAE is a given atomic environment of the target scaffold along with latent vector to generate substructures that are tailored to bond with the scaffold.

 4- A GP model is trained on a labeled dataset that is given using the output of the encoder $z$ from step 3 the latent vector representing a substructure, $p$ the bonding position of the modification and the target is

$y'_{\Delta} = y-y'$

represents the improvement in the property. The GP dataset was prepared as follows: randomly sampling of substructures from random latent vectors $z$, each paired with randomly selected bonding regions $p$ on the scaffold $S$, and evaluated the $y'_\Delta$ obtained from these additions.


5- CLaSMO described in Algorithm 1, searches for optimal substructures while having a similarity
constraint between the original scaffold $S$ with the modified molecule $S'$, the similarity is measured using Dice Similarity while the GP model is used for optimizing the substructure.

### Experiments and Results:
- The paper showed experiments QED Optimization where they showed improvements over the original property up to 11.70% with a high similarity threshold and 21.43% with a no similarity threshold. The paper compared to other models in this task, showing that they can achieve similar QED with a much smaller dataset compared to other models.

- The paper showed experiments Docking Simulation Score Optimization showing mean improvement between 75.1% and 35.4% based on the similarity threshold.

- Ablation studies were performed to compare the effect of using a CVAE vs a VAE.

- Paper mentioned that this method can be used along with a human expert who manually selects the region of the molecule to modify, integrating the expert knowledge into the optimization process, allowing CLaSMO to operate in a Human-in-the-loop setting.

**Audience:**

Yes

**Claims And Evidence:**

Yes

**Requested Changes:**

Some points to clarify:

### Condition Vector Embeddings:
- Why are the Condition Vector Embeddings needed? I understand that you would like to reduce the dimensionality of the 6 features, but 6 is pretty small already, and it seems that if used as is, there probably won't be any significant difference in performance (it actually might improve). No ablation was done to justify this design choice.
 - In section 4.2.1 it was mentioned that an autoencoder was used to obtain Condition Vector Embeddings while in page 10 first paragraph the paper mentioned the following "Prior to CVAE training, we obtained the condition vector embeddings via the encoder of this VAE model, and used them during CVAE training." was a VAE used or an autoencoder?
### CaLMSO:
- How do you "Identify available atoms in the scaffolds for bonding" line 7 algorithm 1? Please clarify this in more detail in the text.

**Strengths And Weaknesses:**

# Strength:
- The paper is well-written and easy to follow.
- The paper addresses an important problem in small molecules, which is that most generative models generate molecules that can not be manufactured by adding similarity constraints; this increases the likelihood of the generated molecule being manufacturable.
- The methods operate in a relatively small data regime.


# Weaknesses:
- Overall the empirical results are not convincing; only one task is compared with other models and in that the performance was on par with others, I do understand that the dataset used to train this model is a lot smaller; however, preparing the dataset to train CLaSMO is quite expensive compared to others and also CLaSMO is probably slower.
- The Human-in-the-loop setting is quite interesting; however, there were no experiments to show that this, in fact, results in any improvements over the original non-human version of CLaSMO.
- CLaSMO has a web application that can be used easily, which is amazing! However, when I tried, it didn't work. I kept on getting "Error running app."
- Some architecture choices were not justifiable; please see Requested Changes.

---

> ### Author Response · Authors · 2025-01-09
> **Added Experiments, CLaSMO Design Choices and Reflection on Comments**
>
> Thank you for your constructive feedback. We acknowledge the concerns raised, particularly regarding the lack of empirical results and design choices of our CVAE model, and have taken steps to address these points. We believe the additional experiments, clarified dataset preparation process, and the now-functional web application addresses the key concerns raised. In the updated manuscript, you will see the revised parts in red.
>
> ## Weakness 1
> > Overall the empirical results are not convincing; only one task is compared with other models and in that the performance was on par with others, I do understand that the dataset used to train this model is a lot smaller; however, preparing the dataset to train CLaSMO is quite expensive compared to others and also CLaSMO is probably slower.
> ### Our Response
> - We have expanded our evaluations with new experiments on 20 tasks, comparing CLaSMO with four other methods. These additional results offer a more comprehensive assessment of CLaSMO’s performance and highlight its strengths across various scenarios.
> - While preparing datasets does require some effort, it is not prohibitively expensive. We included a python notebook containing the data preparation step to the supplementary material, and the entire process finished in 191 seconds, demonstrating that the preparation remains feasible.
> - The main computational cost comes from the Gaussian Process (GP) used in LSBO, which has O(N^3) complexity. Although this can be more intensive than some alternatives, in molecular design, black-box evaluations (e.g., docking or wet-lab tests) are typically far more time-consuming. As a result, focusing on sample efficiency justifies the GP overhead, since it reduces the number of costly experiments needed.
>
> ## Weakness 2
> > The Human-in-the-loop setting is quite interesting; however, there were no experiments to show that this, in fact, results in any improvements over the original non-human version of CLaSMO.
> ### Our Response
> We understand the reviewer’s concerns about the lack of empirical validation for the Human-in-the-Loop (HITL) setting. Our discussion on HITL is motivated by the flexibility that CLaSMO provides to domain experts. Specifically, the framework enables experts to select specific regions of a molecule for modification, which we view as a natural extension of our study. We emphasize that the HITL functionality is not presented as a primary focus of this work but rather as an additional feature to enhance practical usability.
>
> ## Weakness 3
> CLaSMO has a web application that can be used easily, which is amazing! However, when I tried, it didn't work. I kept on getting "Error running app."
> ### Our Response
> We apologize for the inconvenience caused by the web application error. We noticed the same issue, which resolved itself the following day—likely due to a temporary problem with Streamlit’s free servers. The application is now functioning properly, and we welcome any additional feedback on its functionality and the HITL setting.
>
> ## Weakness 4
> > Some architecture choices were not justifiable; please see Requested Changes.
> ### Our Response
>
> We appreciate the reviewer’s interest in our architectural decisions and have provided a detailed explanation in the corresponding section below.
>
> ## Actions for Requested Changes
> ### Change 1
> > Why are the Condition Vector Embeddings needed?
>
> We appreciate the question on Condition Vector Embeddings. Our choice stemmed from using a 2D latent space: if the condition vector exceeds this dimension, the model tends to over-rely on it, leaving the latent space less informative. To compare performance, we trained two CVAEs—one with 2D embeddings of the condition vectors and another with the original 6D features—then extracted each scaffold’s 2D latent representation and trained random forest regressors to predict QED. The 2D embedding version achieved a 5% lower MSE, indicating that its latent vectors provide stronger predictive power. This predictive power is critical for LSBO, where latent vectors feed into the surrogate model that guides exploration. A more informative latent space helps the surrogate model better predict property values, resulting in more efficient searching. However, we acknowledge the 6D model converges faster and has better reconstruction accuracy. In our updated manuscript, we added a more detailed discussion on this to the Appendix A.3.
>
> ### Change 2
> > VAE or Autoencoder when embeddings are obtained?
>
> Thank you for pointing this out and we are sorry for the confusion it caused. We used an autoencoder model, and we updated the part that caused this confusion.
>
> ### Change 3
> > How do you "Identify available atoms in the scaffolds for bonding" line 7 algorithm 1? Please clarify this in more detail in the text.
>
> Thank you for pointing this out. We identify atoms with remaining valence—for example, a carbon with only three bonds—and mark them as “available for bonding.” We clarified this in a footnote on page 10.

---

### Decision · Action_Editor_rEX9 · 2025-01-28

**Recommendation:** Reject

**Comment:**

The method introduces a novel idea of conditional generation based on attachment point, which is further combined with a bayesian optimization algorithm. Reviewers were generally enthusiastic about the proposed method. However, without a more thorough evaluation it is not clear how it compares to other modern approaches.

**Audience:**

The paper would be of interest to the molecular community. However, the shortcomings of the evaluation design would make it less interesting and hinder its broader impact.

**Claims And Evidence:**

The paper proposes a novel method for de novo molecular generation that combines CVAE and (latent) Bayesian optimization.

The main claim is that the method is more sample efficient than alternative methods, in particular due to narrowing the search to more synthetically accessible compounds. However, the evaluation leaves some doubts:
(1) the evaluation showing that it is possible to optimize QED and docking score do not support the main claim due to lack of clear comparison to baselines
(2) comparison on the Gao benchmark (added in the discussion phase) is definitely a significant improvement. However, there were doubts shared by one reviewer regarding hyperparameter selection. Furthermore, methods compared against are not the most recent. In particular, there are methods that focus on fragment-based generation such as FragGFN or SynFlowNet.

From the correctness perspective, I would also suggest that synthetic accessibility is evaluated using a retrosynthesis tool (e.g. AzynthFinder) or similar, as QED is well known to be only a very crude approximation.

**Resubmission Of Major Revision:**

The authors may consider submitting a major revision at a later time.

---

> ### Author Response · Authors · 2025-02-06
> **Reply to Final Decision**
>
> We sincerely appreciate the action editor's comments and the timely decision. We also thank the action editor for assigning knowledgeable reviewers, whose feedback helped us improve our manuscript.
>
> That said, we respectfully disagree with some of the reasons for the rejection. In summary, we find the action editor’s comments inconsistent with our discussions with the reviewers. The action editor’s comments regarding QED experiments, hyperparameters, and benchmark methodologies were already addressed during the discussion period. Given that all reviewers evaluated "Yes" for both the claims and evidence, as well as the audience sections, we find these comments and the final decision unfair.
>
> > the evaluation showing that it is possible to optimize QED and docking score do not support the main claim due to lack of clear comparison to baselines
>
> We had acknowledged this concern in the initial reviews and, in response, expanded our evaluation by adding 20 additional Oracle functions. Our claims are now supported by a comprehensive set of evaluations, not limited to QED and docking scores. Additionally, we clarified the rationale behind the limited evaluation on docking scores (the time-consuming nature of the simulator we use to calculate docking scores), positioning it as a close-to-real-life scenario while supplementing it with other oracle functions. Given these additions, we find this criticism to be inconsistent with the updated manuscript.
>
> > However, there were doubts shared by one reviewer regarding hyperparameter selection.
>
> We carefully addressed the reviewer's concerns by providing full transparency on our hyperparameter selection. The reviewer acknowledged our response and did not raise further objections and increased the evaluation, which we interpreted as a resolution of the issue. Given this, we find it surprising that this point remains a reason for rejection.
>
> > Furthermore, methods compared against are not the most recent. In particular, there are methods that focus on fragment-based generation such as FragGFN or SynFlowNet.
>
> We acknowledge that additional benchmarks exist, and having more benchmark methodologies is always beneficial. Our comparisons were based on the feedback we received from the reviewers, particularly reviewer P2E2. Reviewer P2E2 specifically requested that we include the genetic algorithm and other benchmarks from a widely recognized benchmark study. After incorporating these updates into our manuscript, the reviewer upgraded their evaluation of the Claims and Evidence section from "No" to "Yes." Therefore, we find the action editor’s comments inconsistent with our discussions with the reviewers.
>
> > From the correctness perspective, I would also suggest that synthetic accessibility is evaluated using a retrosynthesis tool (e.g. AzynthFinder) or similar, as QED is well known to be only a very crude approximation.
>
> We completely agree with the criticism regarding synthetic accessibility and have no objections. However, as discussed above, we had already acknowledged the issue with QED and addressed it by adding 20 additional benchmark oracles to our evaluations. Therefore, given the updated manuscript, we find the inclusion of QED usage as a reason for rejection unfair.
>
> We appreciate the constructive feedback from both the action editor and the reviewers, and of course, we acknowledge that further refinements could strengthen our work.
>
> Best regards

---

> > ### Comment · Action_Editor_rEX9 · 2025-02-06
> > **Thank you**
> >
> > Thank you for your comment. I know it is a dissapointing outcome and I am sorry to hear that you feel the decision was not fair.
> >
> > I wanted to remark (and that should have been mentioned in the meta review) that in the end one reviewer voted for rejection, mentioning in reasoning doubts about hyperparameter choice.
> >
> > The paper is welcomed for resubmission provided the issues are fully adressed.

---

> > > ### Author Response · Authors · 2025-02-11
> > > **Updates for Resubmission**
> > >
> > > Dear Action Editor,
> > >
> > > We sincerely thank you for receiving our message positively and for encouraging us to resubmit.
> > >
> > > We would like to resubmit, but find it difficult to determine where and how to update the paper. It appears that all reviewers are satisfied with our revision and have not pointed out any further issues (we believe you will agree if you review the discussion).
> > > Regarding the hyperparameter selection mentioned in your reply, Reviewer P2E2 expressed concerns initially but was satisfied with our revision and changed her/his evaluation positively (we have received no comments regarding hyperparameters from other reviewers).
> > >
> > > For resubmission, we would appreciate it if the action editor or the reviewers could specify what is the remaining concerns with hyperparameter selection.
> > >
> > > We understand that TMLR editorial policy emphasizes technical correctness over subjective significance. If the paper is rejected for reasons not pointed out in the review, we are afraid this may not be consistent with the policy.
> > >
> > > Thank you again for dedicating your precious time to our paper.
> > >
> > > Best regards